# HadamRNN: Binary and sparse ternary Orthogonal RNNs

**Armand Foucault**
Institut de Mathématiques de Toulouse,
UMR5219. Université de Toulouse, CNRS. UPS IMT,
F-31062 Toulouse Cedex 9, France
`armand.foucault@math.univ-toulouse.fr`

**Franck Mamalet**
Institut de Recherche Technologique Saint Exupéry
Toulouse, France
`franck.mamalet@irt-saintexupery.com`

**François Malgouyres**
Institut de Mathématiques de Toulouse,
UMR5219. Université de Toulouse, CNRS. UPS IMT,
F-31062 Toulouse Cedex 9, France
`francois.malgouyres@math.univ-toulouse.fr`

## Abstract

Binary and sparse ternary weights in neural networks enable faster computations and lighter representations, facilitating their use on edge devices with limited computational power. Meanwhile, vanilla RNNs are highly sensitive to changes in their recurrent weights, making the binarization and ternarization of these weights inherently challenging. To date, no method has successfully achieved binarization or ternarization of vanilla RNN weights. We present a new approach leveraging the properties of Hadamard matrices to parameterize a subset of binary and sparse ternary orthogonal matrices. This method enables the training of orthogonal RNNs (ORNNs) with binary and sparse ternary recurrent weights, effectively creating a specific class of binary and sparse ternary vanilla RNNs. The resulting ORNNs, named HadamRNN and Block-HadamRNN, are evaluated on various benchmarks, including the copy task, permuted and sequential MNIST tasks, the IMDB dataset, two GLUE benchmarks, and two IoT benchmarks. Despite binarization or sparse ternarization, these RNNs maintain performance levels comparable to state-of-the-art full-precision models, highlighting the effectiveness of our approach. Notably, our approach is the first solution with binary recurrent weights capable of tackling the copy task over 1000 timesteps.

## 1 Introduction

A Recurrent Neural Network (RNN) is a neural network architecture relying on a recurrent computation mechanism at its core. These networks are well-suited for the processing of time series, thanks to their ability to model temporal dependencies within data sequences. Traditional Recurrent architectures such as vanilla RNNs, LSTM (Hochreiter and Schmidhuber, 1997), GRU (Cho et al., 2014) or Unitary/Orthogonal RNN (Arjovsky et al., 2016; Helfrich et al., 2018) have achieved remarkable performance across various sequential tasks including neural machine translation (Devlin et al., 2014; Sutskever et al., 2014) and speech recognition (Amodei et al., 2016; Chan et al., 2016).

Modern RNN architectures typically rely on millions, or even billions, of parameters to perform optimally. This necessitates substantial storage space and costly matrix-vector products at inference-time, that may result in computational delays. These features can be prohibitive when applications must operate in real-time or on edge devices with limited computational resources.

A compelling strategy to alleviate this problem is to replace the full-precision weights within the network with weights having a low-bit representation. This strategy, known as neural network quantization (Courbariaux et al., 2015; Lin et al., 2015; Courbariaux et al., 2016; Hubara et al., 2017; Zhou et al., 2016), has been extensively studied over the recent years. For optimal computational efficiency and memory savings, weights should be binarized, that is, represented over only 1 bit. For

the case of recurrent networks, it was shown (Ott et al., 2016; He et al., 2016; Hou et al., 2017; Alom et al., 2018; Ardakani et al., 2019) that LSTMs and GRUs with binarized weights could achieve near state-of-the-art results on natural language datasets such as Penn TreeBank (Marcus et al., 1993), Leo Tolstoy's *War and Peace*, (Hou et al., 2017) or IMDB (Maas et al., 2011).

However, vanilla RNNs, LSTMs and GRUs usually struggle at learning tasks involving very long-term dependencies, notably due to the exploding gradient problem (Pascanu et al., 2013). In Arjovsky et al. (2016); Wisdom et al. (2016); Lezcano-Casado et al. (2019) for instance, it can be observed that LSTMs fail to solve the copy task with long sequences. In recent years, alternative models including transformers (Vaswani et al., 2017), ODE-inspired RNNs (Erichson et al., 2021; Rusch and Mishra, 2021) including SSMs (Gu et al., 2022) were designed to accurately model long-term dependencies. Among these, recent studies have tackled the challenge of quantizing transformers, but have only achieved model sizes in the tens of megabytes (see Section 2 and Appendix F.1). The only known attempt to quantize SSMs reports a significant loss of accuracy when reducing precision below 8-bit (Abreu et al., 2024). Hence the need for efficient, lightweight, binary recurrent architectures capable of handling longer dependencies than LSTMs and GRUs.

**Contribution:** In this paper, we binarize the recurrent weights of Orthogonal Recurrent Neural Networks (ORNNs), which is a special case of vanilla RNNs. The binary orthogonal matrices are constructed using Hadamard matrix theory. We call these networks Hadamard RNNs (HadamRNN). To the best of our knowledge, this is the first successful attempt to binarize the weights of vanilla and orthogonal RNNs. To reduce the complexity further, we also build sparse ternary ORNNs called Block-Hadamard RNNs (Block-HadamRNN). The resulting HadamRNNs and Block-HadamRNNs are fully-quantized, lightweight, highly efficient, and model long-term dependencies more accurately than LSTMs and GRUs. This claim is supported by results on a variety of benchmarks including the copy task for 1000 timesteps, permuted MNIST, pixel-by-pixel MNIST, the IMDB dataset, two GLUE benchmarks, and two IoT benchmarks. Despite the drastic reduction in computational complexity and memory footprint, the performance degradation remains moderate compared to full-precision ORNNs. Ablation studies show the effectiveness of the choices made in the article.

**Organization of the paper** In Section 2, we review previous work on binarizing and ternarizing the weights of neural networks for sequential data modeling. The bibliography is supplemented by Appendix A, which focuses on ORNNs. In Section 3, we describe the method for parameterizing a subset of binary and sparse ternary orthogonal matrices. We also outline all the components of HadamRNNs and Block-HadamRNNs. Experiments are detailed in Section 4, and conclusions are provided in Section 5.

Additional bibliographic references are in Appendix A, and proofs are provided in Appendix B. Appendix C illustrates the model description. Details for reproducing the experiments, along with additional experimental results, are in Appendix D, Appendix E, and Appendix F. Theoretical complements and an ablation study on the (linear) ORNN architecture are in Appendix G, while those on the proposed parameterization of orthogonal matrices are in Appendix H. The fixed-point arithmetic implementation is detailed in Appendix I.

The code implementing the experiments is available at hadamRNN

## 2 Related works

This section provides an overview of prior research efforts aimed at quantizing the weights of neural networks designed for handling sequential data, encompassing both recurrent and non-recurrent architectures. We first highlight the works that report performance with binary and ternary recurrent weights. Further bibliographical details on ORNNs can be found in Appendix A.

In the seminal paper Ott et al. (2016), the authors apply binarization and ternarization methods on vanilla RNN, LSTM and GRU architectures. Remarkably, they acknowledge the difficulty of training binary RNNs; they write at the beginning of Section 3.1.1: 'We have observed among all the three RNN architectures that BinaryConnect on the recurrent weights never worked.'In Hou et al. (2017), the authors apply a loss-aware binarization scheme to an LSTM and achieve better performances than the conventional BinaryConnect algorithm (Courbariaux et al., 2015) on a language modeling task. Using a learning process incorporating stochasticity and batch-normalization, Ardakani et al. (2019) show that an LSTM and a GRU with binary weights can achieve results comparable to their

full-precision counterparts on language modeling tasks and the sequential MNIST task. Finally, He et al. (2016); Liu et al. (2018); Wang et al. (2018); Alom et al. (2018) take a step further toward a fully quantized recurrent network. He et al. (2016) proposes to quantize the activations of a GRU and an LSTM in addition to binary weights. Liu et al. (2018) suggests to binarize the word embeddings as inputs for an LSTM with binary weights. Alom et al. (2018) propose another quantization scheme and implement quantized convolutional LSTM which are tested on the moving MNIST dataset.

Among the articles cited above, Ott et al. (2016); Hou et al. (2017); Ardakani et al. (2019); He et al. (2016) consider character-level language models. The works by Ardakani et al. (2019); Wang et al. (2018); Liu et al. (2018) report performance results for word-level language models. He et al. (2016); Alom et al. (2018) address sentiment analysis on the IMDB dataset (Maas et al., 2011). Ardakani et al. (2019) is the only article reporting results for a long-term dependency problem, that LSTMs are known to solve efficiently: the sequential MNIST problem. Kusupati et al. (2018) is the only work to solve IoT tasks. None of the articles attempts to solve the permuted MNIST problem or the copy task for 1000 timesteps, both of which are known to be better addressed by ORNNs.

Quantization methods for LSTMs and GRUs using larger bit-widths have been described in Hubara et al. (2017); Kusupati et al. (2018); Nia et al. (2023); Xu et al. (2018); Zhou et al. (2017).

To the best of our knowledge, the only article describing a method for quantizing ORNNs is Foucault et al. (2024). In this article, the authors succeed in learning tasks involving long-term dependency with a 4-bit ORNN.

We classify existing efforts to quantize Transformers such as BERT (Devlin, 2018) based on the level of quantization applied. Some approaches focus on quantizing the weights to 8 bits (Zafrir et al., 2019; Sun et al., 2020; Stock et al., 2021) or even 4 bits (Shen et al., 2020; Zadeh et al., 2020). Others explore more aggressive quantization, employing ternary weights (Zhang et al., 2020) or binary weights (Bai et al., 2021). Fully binarized versions of BERT, including binarized activations, are presented in (Qin et al., 2022; Liu et al., 2022).

As shown in Appendix F.1, Table 6, all these networks require at least tens of megabytes of storage. This contrasts sharply with the models described in this article, which require much smaller sizes.

Finally, Yao et al. (2022); Frantar et al. (2023); Xiao et al. (2023); Liu et al. (2023) extend quantization techniques to large language models with billions of parameters.

# 3 HADAMARD AND BLOCK-HADAMARD RNNs

We describe the details of the considered ORNNs in Section 3.1. A brief review of the key properties of Hadamard matrices is provided in Section 3.2. We explain, in Section 3.3, how Hadamard matrices are used to build ORNNs with binary recurrent weights that we call HadamRNN. We extend the construction to sparse ternary recurrent weight matrices, referred to as Block-HadamRNN, in Section 3.4. We describe how input and output weight matrices are quantized in Section 3.5 and compare the complexities of the proposed models in Section 3.6.

## 3.1 ORNNs

Orthogonal recurrent networks are a class of recurrent networks that rely on the same recurrent operation as the one of a vanilla recurrent network, but add an orthogonality constraint on the recurrent weight matrix. Given a sequence of inputs $x_1, \ldots, x_T \in \mathbb{R}^{d_{in}}$, the model computes a sequence of hidden states $h_1, \ldots, h_T \in \mathbb{R}^{d_h}$ according to

$$h_t = W h_{t-1} + U x_t + b_i, \tag{1}$$

where $h_0 = 0$, a matrix $U \in \mathbb{R}^{d_h \times d_{in}}$, $b_i \in \mathbb{R}^{d_h}$, and the recurrent weight matrix $W \in \mathbb{R}^{d_h \times d_h}$ is constrained to be orthogonal (i.e. $W'W = WW' = I_{d_h}$, where $W'$ is the transpose of $W$, and $I_{d_h}$ is the identity matrix of size $d_h \times d_h$). Depending on the task, the output is either the vector $V\sigma(h_T) + b_o \in \mathbb{R}^{d_{out}}$ or the time series $V\sigma(h_1) + b_o, \ldots, V\sigma(h_T) + b_o$. The matrix $V \in \mathbb{R}^{d_{out} \times d_h}$ is the output matrix, and $\sigma$ is the activation function.

The orthogonality of the recurrent weight matrix enhances memorization and prevents gradient vanishing. These networks have been shown to solve complex tasks with long-term dependencies,

such as the copy task with 1000 timesteps or more Lezcano-Casado et al. (2019); Helfrich et al. (2018); Vorontsov et al. (2017); Mhammedi et al. (2017). They also lead to simple RNNs, whose inference complexity scales linearly with sequence length.

The most common choice in ORNNs is to apply the ReLU activation function, $\sigma$, to each hidden state update. The formula then becomes $h_t = \sigma(W h_{t-1} + U x_t + b_i)$, and the output is simply $V h_T + b_o$. In (1), we consider ORNNs with linear recurrent units. The use of the linear recurrent unit improves memorization (see Appendix G) and is also motivated by studies on SSMs (Gu et al., 2022; Orvieto et al., 2023). We provide an ablation study to support this choice in Appendix G.1.

### 3.2 Introduction to Hadamard matrices theory

Before describing how we construct binary or sparse ternary recurrent weight matrices in Sections 3.3 and 3.4, we first recall known properties of Hadamard matrices (Hedayat and Wallis, 1978), and explain how, under simple conditions, we can parameterize a subset of all Hadamard matrices.

**Definition 3.1.** Hadamard matrices (Hadamard, 1893) are square matrices with binary values in $\{-1, 1\}$, whose rows are pairwise orthogonal. For any $n \in \mathbb{N}^*$, we denote by $\mathcal{H}_n$ the (possibly empty) set of all Hadamard matrices of size $n \times n$.

Notice that for any $n > 1$ and any Hadamard matrix $W$ of size $n \times n$, we have

$$W W' = n I_n. \tag{2}$$

It is well known that for $n > 2$, Hadamard matrices of size $n \times n$ do not exist unless $n$ is a multiple of 4 (Hedayat and Wallis, 1978). The existence of Hadamard matrices of size $4n \times 4n$ for all $n > 1$ remains a conjecture. It is called the Hadamard conjecture (De Launey and Gordon, 2001). It is therefore hopeless to attempt learning an optimal matrix in $\mathcal{H}_n$ for an arbitrary $n$.

The following proposition outlines a straightforward method, introduced in Sylvester (1867), to construct a Hadamard matrix of size $2^k \times 2^k$ for any $k \geq 1$.

**Proposition 3.2.** *Let $k \geq 1$. The $2^k \times 2^k$ matrix, denoted $\mathbf{S}_{2^k}$, defined recursively by*

$$\mathbf{S}_2 = \begin{pmatrix} 1 & 1 \\ 1 & -1 \end{pmatrix} \quad (i.e.\ if\ k = 1), \quad and \quad \mathbf{S}_{2^k} = \begin{pmatrix} \mathbf{S}_{2^{k-1}} & \mathbf{S}_{2^{k-1}} \\ \mathbf{S}_{2^{k-1}} & -\mathbf{S}_{2^{k-1}} \end{pmatrix} \quad , if\ k > 1,$$

*is a Hadamard matrix. It is called the Sylvester matrix[1] of size $2^k$ (Horadam, 2007).*

The proof is provided for completeness in Appendix B.2.

For any $n > 1$, if a Hadamard matrix of size $n \times n$ is known, the following proposition provides a simple method for generating $2^n$ distinct Hadamard matrices. In the proposition, the notation $\mathrm{diag}(u) \in \mathbb{R}^{n \times n}$ refers to a diagonal matrix with $u \in \mathbb{R}^n$ on its diagonal.

**Proposition 3.3.** *For $n > 1$ and any $H \in \mathcal{H}_n$, the mapping*

$$\phi_H : \{-1, 1\}^n \longrightarrow \{-1, 1\}^{n \times n}$$
$$u \longmapsto \mathrm{diag}(u) H,$$

*is injective. Moreover, for all $u \in \{-1, 1\}^n$, $\phi_H(u)$ is a Hadamard matrix.*

The proof is provided for completeness in Appendix B.3. This proposition guarantees that switching the signs of any set of rows of a Hadamarad matrix preserves its Hadamard property. Considering the matrices $\phi_{\mathbf{S}_{2^k}}(u)$, for $u \in \{-1, 1\}^{d_h}$, Proposition 3.2 and Proposition 3.3 provide a method for manipulating $2^{2^k}$ Hadamard matrices of size $2^k \times 2^k$, for $k \geq 1$. For instance, when $d_h = 2^8 = 256$, this allows the generation of more than $10^{77}$ different matrices. The experiments will confirm that this class of matrices possesses sufficient expressiveness.

Note that even if, in the following, we only use Proposition 3.3 with Sylvester matrices whose size is a power of 2, Proposition 3.3 applies to any given Hadamard matrix $H$. Many such matrices exist.

---

[1]These matrices are also called Walsh matrices in some contexts.

In particular, Hadamard matrices can be constructed for almost all sizes $4n \times 4n$, when $4n \leq 2000$ (Đoković et al., 2014). This would allow parameter $d_h$ to be set more finely than we have done.

A key point is that, due to the independence of $u$'s components, empirical results show that $u$ can be optimized using standard methods like the straight-through estimator (STE) (Hinton, 2012; Courbariaux et al., 2016), as described in Appendix C.

Finally, similarly to what has been done in Proposition 3.3, it can also be shown that it is possible to switch the signs of any set of columns of a Hadamard matrix and preserve the Hadamard property. We argue in Appendix H.1 that, because we also optimize the input matrix $U$ and the input bias $b_i$, it does not lead to more expressive networks. Additionally, the ablation study in Appendix H.2 shows that it does not allow for any improvement in practice.

### 3.3 BINARY ORTHOGONAL RECURRENT WEIGHT MATRICES

To parameterize the binary orthogonal recurrent weights used in our network, reffered to as Hadamard RNN (HadamRNN), we consider $d_h = 2^k$, for $k \geq 1$, and the weights

$$W(u) = \frac{1}{\sqrt{d_h}} \operatorname{diag}(u) \mathbf{S}_{2^k} \in \mathbb{R}^{d_h \times d_h}, \tag{3}$$

for a trainable binary vector $u \in \{-1, 1\}^{d_h}$. Indeed, using that $\operatorname{diag}(u) \mathbf{S}_{2^k}$ is a Hadamard matrix satisfying (2), we obtain that $W(u)$ is orthogonal: $W(u)'W(u) = W(u)W(u)' = I_{d_h}$. The proof is provided in Appendix B.4. We also detail an example in Appendix C. It is worth noting that if $k$ is even, $d_h = 2^{2k'}$, for $k' \geq 1$, then the normalization becomes a division by $\sqrt{d_h} = 2^{k'}$, which is well-suited for efficient implementation on edge devices.

### 3.4 SPARSE TERNARY ORTHOGONAL RECURRENT WEIGHT MATRICES

To construct the sparse ternary orthogonal recurrent weights used in Block-Hadamard RNNs (Block-HadamRNNs), we consider $d_h = q2^k$, where $k \geq 1$ and $q \geq 1$, with the weights defined as

$$W(u) = \frac{1}{\sqrt{2^k}} \operatorname{diag}(u)\left(I_q \otimes \mathbf{S}_{2^k}\right) \in \mathbb{R}^{d_h \times d_h}, \tag{4}$$

for a trainable binary vector $u \in \{-1, 1\}^{d_h}$ and the Kronecker product $\otimes$ (see Appendix B.1). The matrix $W(u)$ is ternary since its components are in $\left\{-\frac{1}{\sqrt{2^k}}, 0, \frac{1}{\sqrt{2^k}}\right\}$. It is orthogonal for the same reasons outlined in the previous section. The proof is detailed in Appendix B.4.

The proportion of non-zero components in $W(u)$ is $\frac{q(2^k)^2}{(q2^k)^2} = \frac{1}{q}$. When $q$ is large, the matrix $W(u)$ is very sparse. On the contrary, when $q = 1$, none of the components of $W(u)$ is zero, and the Block-Hadamard RNN effectively becomes a Hadamard RNN as described in the previous section. In this sense, Block-Hadamard RNNs are a natural sparse ternary extension of Hadamard RNNs.

### 3.5 MATRICES $U$ AND $V$ QUANTIZATION

Because input and output sizes are often much smaller than the size of the hidden space (i.e. $d_{in} \ll d_h$ and $d_{out} \ll d_h$), we permit the quantization of the input and output weight matrices, $U$ and $V$, using $p$ bits, where $p \geq 2$. We use the uniform quantization with a scaling parameter (Gholami et al., 2022).

The quantization approximates every component of $U$ (resp. $V$) by its nearest element in the set

$$\frac{\alpha}{2^{p-1}} \left[\!\left[ -2^{p-1}, 2^{p-1} - 1 \right]\!\right].$$

where $\alpha = \max_{i,j} |U_{ij}|$ (resp. $\alpha = \max_{i,j} |V_{ij}|$) and the set $[\![a, b]\!]$ contains all the integers between $a$ and $b$. The above set contains $2^p$ elements.

To obtain ternary $U$ and $V$, leading to matrix-vector multiplications involving only additions, we also provide the results for the quantization approximating each component of $U$ (resp $V$) in $\alpha\{-1, 0, 1\}$, for the same values of $\alpha$.

Moreover, when the input is one-hot encoded, only one column of $U$ is used at a time. The product $U x_t$ can take only $d_{in}$ values and, from a computational perspective, can be encoded as the input bias $b_i$ and the activation.

### 3.6 MODEL SIZE AND COMPUTATIONAL COMPLEXITY

We compare the different models in terms of parameter storage requirements and the number of operations during inference.

The model size is determined by the total number of learnable parameters, multiplied by the number of bits used to encode each parameter. In both HadamRNNs and Block-HadamRNNs, the recurrent layer requires $d_h$ bits for encoding the binary vector $u$, the input matrix $U$ requires $d_{in}d_h p$ bits, and the output matrix $V$ requires $d_h d_o p$ bits. Therefore, the total model size is given by $\frac{d_h(1+(d_{in}+d_o)p)}{8\times1024}$ kBytes[2]. It is important to note that using Sylvester matrices $\mathbf{S}_{2^k}$ eliminates the need to store their weights, as these can be easily retrieved using Proposition 3.2.

The number of operations of the inference using HadamRNN and Block-HadamRNN is detailed in Table 1. We assume in Table 1 that the hidden and input variables, $h_t$ and $x_t$, are encoded using $p_a$ bits. A detailed description of the fully quantized RNN operations is given in Appendix I. HadamRNNs and Block-HadamRNNs use binary (or ternary) recurrent matrices, which eliminates the need for multiplications. Similarly, for ternary $U$ and $V$, we set $\mathrm{fpp}_{p,p'} = 0$ since the matrix-vector multiplications only involve additions. When $p = 2$, the matrix-vector multiplications involving matrices $U$ and $V$ in $\{-2, -1, 0, 1\}$ only involve additions and bit-shifts. For the same $d_h$ value ($d_h = 2^k = q.2^{k'}$ with $k' < k$), the computational complexity of the recurrent layer of Block-HadamRNN is $q$ times lower than that of HadamRNN.

For comparison, we also provide the complexity for the inference with full-precision ORNN (Arjovsky et al., 2016) and the only quantized ORNN that we are aware of: QORNN (Foucault et al., 2024). The complexities of HadamRNN and Block-HadamRNN are much smaller, in particular, because, as will be reported in Section 4, they permit to achieve satisfactory results for $p = 2$ when, as reported in Foucault et al. (2024), QORNNs require at least $p = 4$ bits encoding.

Note that when the inputs $x_t$ are one-hot encoded, computing $U x_t$ requires no multiplications and only $d_h$ additions. This further reduces the complexity compared to Table 1.

Table 1: Computational complexity for an inference of the RNN. We neglect the bit-shifts and the accesses to the look-up tables. FP stands for in floating-point arithmetic, fpp stands for fixed-point precision additions, $\mathrm{fpp}_{p,p_a}$ stands for fixed-point precision multiplications between numbers coded using $p$ and $p_a$ bits. We have $\mathrm{fpp}_{t,p_a} = \mathrm{fpp}_{2,p_a} = 0$, where $\mathrm{fpp}_{t,p_a}$ is for ternary matrices.

| Layer | Operation | ORNN | QORNN | HadamRNN $d_h = 2^k$ | Block-HadamRNN $d_h = q2^k$ |
|---|---|---|---|---|---|
| Input | Mult. | $d_{in}.d_h$ FP | $d_{in}.d_h$ $\mathrm{fpp}_{p,p_a}$ | idem | idem |
| | Add. | $d_{in}.d_h$ FP | $d_{in}.d_h$ $\mathrm{fpp}$ | idem | idem |
| Recurrent | Mult. | $d_h.d_h$ FP | $d_h.d_h$ $\mathrm{fpp}_{p,p_a}$ | 0 | 0 |
| | Add. | $d_h.d_h$ FP | $d_h.d_h$ $\mathrm{fpp}$ | $d_h.d_h$ $\mathrm{fpp}$ | $d_h.\frac{d_h}{q}$ $\mathrm{fpp}$ |
| Output | Mult. | $d_h.d_{out}$ FP | $d_h.d_{out}$ $\mathrm{fpp}_{p,p_a}$ | idem | idem |
| | Add. | $d_h.d_{out}$ FP | $d_h.d_{out}$ $\mathrm{fpp}$ | idem | idem |

## 4 EXPERIMENTS

In this section, we assess the performance of HadamRNN and Block-HadamRNN on four standard benchmark datasets. These datasets are described in Section 4.1. Three tasks require retaining

---

[2]The biases $b_i$ and $b_o$ use $(d_h + d_o)p_a$ bits.

information over extended periods, while the fourth focuses on a Natural Language Processing (NLP) task with shorter sequences but larger input dimensions. In Section 4.2, we present the performance of HadamRNN and Block-HadamRNN and compare them to that of previously published quantized and full-precision (FP) models. We conduct ablation studies in Appendix G.1 and Appendix H.2.

## 4.1 DATASETS

We investigate lightweight neural networks for time series and select datasets suited to these architectures. In particular, this excludes the Long Range Arena benchmarks (Tay et al., 2020), which are too complex for full-precision ORNNs, LSTMs, and GRUs. To illustrate the limitations of the proposed models, we include the sequential MNIST for which LSTMs are known to outperform ORNNs. **Copy task** The Copy task is a standard sequential problem first introduced in (Hochreiter and Schmidhuber, 1997). This task requires memorizing information over many timesteps, and vanilla LSTMs are notoriously unable to solve it for long sequences (Arjovsky et al., 2016; Helfrich et al., 2018; Lezcano-Casado et al., 2019). We follow the setup of Lezcano-Casado et al. (2019), in which the data sequences are constructed as follows. We consider an alphabet $\{a_k\}_{k=0}^{9}$ of 10 characters. Given a sentence length $K$ and a delay $L$, the first $K$ elements of an input sequence are sampled uniformly and independently from $\{a_k\}_{k=1}^{8}$. These are followed by $L$ repetitions of the *blank* character $a_0$, one instance of the *marker* $a_9$, and $K - 1$ repetitions of $a_0$. The first $K$ elements form a sentence that the network must memorize and reproduce identically after outputting $L + K$ instances of $a_0$.

In our experiments, we fixed $K = 10$ and $L = 1000$ ($T = L + 2K = 1020$, $d_{in} = 10$, $d_{out} = 9$). The loss function is the cross-entropy, which is also used to measure performance. A naive baseline consists of $L + K$ repetitions of $a_0$, followed by $K$ random values. This leads to a baseline cross-entropy of $\frac{10 \log 8}{L + 2K} = 0.021$.

**Permuted and sequential pixel-by-pixel MNIST (pMNIST/sMNIST)** They are also classic long-term memory tasks. From the MNIST dataset, the $28 \times 28$ images are serialized into 784-long sequences of 1-dimensional 8-bits pixel values ($T = 784$, $d_{in} = 1$, $d_{out} = 10$). The serialization is done pixel-by-pixel for sMNIST. For pMNIST, a fixed permutation is used to shuffle the pixels within each sequence. We apply the same permutation as Kiani et al. (2022). The task is to predict the correct handwritten digit label at the last step. The learning loss is the cross-entropy, and the model's performance is evaluated with accuracy.

**IMDB** This dataset, proposed in Maas et al. (2011), is an NLP binary classification task for sentiment analysis based on 50,000 movie reviews. As in He et al. (2016), we pad and cut the sentences to 500 words, and use a learnable word embedding vector of size 512 ($T = 500$, $d_{in} = 512$, $d_{out} = 1$). The learning loss is the binary cross-entropy, and the model's performance is evaluated with accuracy.

**Other Datasets** We provide in Appendix F a comparison of HadamRNN, Block-HadamRNN, and transformers on the SST-2 and QQP benchmarks from GLUE (Wang et al., 2019), as well as with RNNs on IoT task benchmarks: HAR-2, and DSA-19, as described in Kusupati et al. (2018).

## 4.2 PERFORMANCE EVALUATION

The evaluation is organized as follows. In Section 4.2.1 and Table 2, we compare the results of HadamRNN to those of the state-of-the-art. In Section 4.2.2 and Table 3, we compare the results of Block-HadamRNN and HadamRNN.

For each task, hyperparameters were selected using validation sets, and final performance was evaluated on test sets. Details on the hyperparameters and implementations are provided in Appendix D.

Training times are provided in Appendix E.1. Training stability is analyzed in Appendix E.2.

### 4.2.1 HADAMRNN VERSUS THE STATE-OF-THE-ART

Since we aim to design high-performance RNN architectures adapted to low-memory devices, we assess the models' performance based on two criteria: the model size of each architecture and its classification accuracy or cross-entropy for the copy task. The model size of HadamRNN is calculated

as described in Section 3.6. LSTM recurrent matrix is the identity matrix and is coded using 0 bits. We also report the performance of fully quantized HadamRNN using the post-training quantization strategy for activations detailed in Appendix I. The main performance results on the four benchmark datasets are summarized in Table 2. The plots in Appendix E.3, based on the results from Table 2, highlight the efficiency of HadamRNN in terms of model size.

**State-of-the-art** We compare HadamRNN with state-of-the-art full-precision and quantized models designed for time series modeling:

- Full-precision LSTM (Jing et al., 2017; Kiani et al., 2022) and quantized LSTM (Ardakani et al., 2019; Alom et al., 2018; He et al., 2016) are known to fail the 1000-timesteps Copy task, and to be well-suited for modeling the sMNIST problem. LSTM also serves as an optimistic proxy for other gated models such as GRU, since LSTM is known to usually perform better.

- Similarly to HadamRNN, ORNN (Kiani et al., 2022) and QORNN (Foucault et al., 2024) are instances of RNNs with orthogonal recurrent weight matrix, but operating with different bitwidth. This comparison evaluates the performance degradation caused by binarization.

- FastGRNN (Kusupati et al., 2018) was designed to address similar problems as unitary RNNs (Arjovsky et al., 2016; Helfrich et al., 2018), but with significantly smaller size. Since HadamRNN is also remarkably lightweight, it appears relevant to compare the two models.

- As stated in Section 1, quantized SSMs perform poorly when the bitwidth is smaller than 8 bits (Abreu et al., 2024). Also, as discussed in Section 2 and Appendix F.1, the differences in size and complexity between transformers and HadamRNN render the comparison meaningless. For these reasons, we do not include the results of transformers and SSMs.

**Copy-task** Notably, HadamRNN is the first binary recurrent weights solution capable of learning the long-term dependency of the Copy task. Note that full-precision and quantized LSTMs do not learn the Copy task when $T = 1020$. The proposed HadamRNN outperforms the 5-bits QORNN introduced in Foucault et al. (2024), while requiring a smaller recurrent size $d_h$. The resulting model size reduction is over 36 (= 50.6/1.40). This improvement stems from the orthogonal nature of the Hadamard matrices, which enables better learning of long-term dependencies compared to QORNNs, where the matrices are only approximately orthogonal. This is also attributed to the choice of a linear recurrent unit, which enhances memorization (see Appendix G).

**pMNIST** Similarly, on pMNIST, the fully quantized HadamRNN, with a size of just 3.58 kB, outperforms the QORNN of Foucault et al. (2024), which requires 35 kB, and even a full-precision ORNN of 120 kB. Additionally, the HadamRNN model of 6.2 kB achieves only 1.2% lower accuracy compared to an ORNN with $d_h = 512$ of much larger size, 1046 kB. All HadamRNN architectures, except for the smallest one, outperform full-precision LSTMs.

**sMNIST** For this task, which typically favors gated models like LSTM and FastGRNN, the Hadam-RNN model of size 3.58kB achieves an accuracy that is only 2% lower compared to LSTMs and FastGRNNs of the same size, and a full-precision ORNN of size 1046kB. It still outperforms the QORNN, achieving 96.9% accuracy with a model size that is 5.6 (= 35/6.23) times smaller.

**IMDB** The smallest HadamRNN models with $d_h = 128$ outperform Alom et al. (2018) binary LSTM, while being 2.4 (= 40.5/16.55) times smaller, and when increased to $d_h = 512$ with a size of 129 kB, it achieves 10% higher accuracy compared to the 40.5 kB model. It also outperforms the full precision ORNN with $d_h = 128$. The 2-bits (resp. 4-bits) LSTM proposed in He et al. (2016) only offers 1% accuracy advantage over HadamRNN at the same recurrent size $d_h = 512$ but their model size is 4 (resp. 8) times larger.

**Activation quantization** Regardless of the task, the results of Table 2 demonstrate that post-training quantization of activations, as described in Appendix I, does not degrade performance.

### 4.2.2 BLOCK-HADAMRNN VERSUS HADAMRNN

When Block-HadamRNN and HadamRNN share the same hidden-space dimension, they have equal size. Their main difference lies in the computational complexity of their recurrent unit, which, as explained in Section 3.6 and Table 1, is $q$ times lower for Block-HadamRNN. In Table 3, we therefore report the computational complexity and the performance for the four considered tasks.

Table 2: Comparison of HadamRNN and the state-of-the-art on several benchmark datasets. Last column reports the model size in kBytes. BL (baseline) means that the model failed to learn.

| Model | $d_h$ | W bitwidth | U & V bitwidth | activation bitwidth | performance | size kBytes |
|---|---|---|---|---|---|---|
| **Copy task** ($T = 1000$, $d_{in} = 10$, $d_{out} = 9$, cross-ent, baseline = 0.021) | | | | | | |
| LSTM (Jing et al. (2017)) | 80 | FP | FP | FP | BL | 112 |
| ORNN (Kiani et al. (2022)) | 256 | FP | FP | FP | 1.1e-12 | 275 |
| QORNN (Foucault et al. (2024)) | " | 8 | 8 | 12 | 1.7e-5 | 75.5 |
| QORNN (Foucault et al. (2024)) | " | 5 | 5 | 12 | 2.5e-3 | 50.6 |
| | | | 2 | FP | BL | 1.44 |
| | | | 4 | FP | 1.6e-7 | 1.74 |
| HadamRNN (ours) | 128 | 1 | 4 | 12 | 2.3e-7 | 1.40 |
| | | | 6 | FP | 3.2e-8 | 2.33 |
| **Permuted MNIST** ($T = 784$, $d_{in} = 1$, $d_{out} = 10$, accuracy) | | | | | | |
| ORNN (Kiani et al. (2022)) | 512 | FP | FP | FP | 97.00 | 1046.0 |
| ORNN (Kiani et al. (2022)) | 170 | FP | FP | FP | 94.30 | 120.2 |
| LSTM (Kiani et al. (2022)) | " | FP | FP | FP | 92.00 | 456.9 |
| QORNN (Foucault et al. (2024)) | " | 8 | 8 | 12 | 94.76 | 35.0 |
| QORNN (Foucault et al. (2024)) | " | 6 | 6 | 12 | 93.94 | 27.9 |
| | | | 2 | FP | 91.13 | 3.48 |
| | | | 4 | FP | 94.88 | 4.85 |
| HadamRNN (ours) | 512 | 1 | 4 | 12 | 94.90 | 3.58 |
| | | | 6 | FP | 95.85 | 6.23 |
| **Sequential MNIST** ($T = 784$, $d_{in} = 1$, $d_{out} = 10$, accuracy) | | | | | | |
| LSTM (Ardakani et al. (2019)) | 100 | 0 | 1 | 12 | 98.6 | 5.11 |
| FastGRNN (Kusupati et al. (2018)) | 170 | 8 | 8 | 16 | 98.2 | 6.0 |
| ORNN (Lezcano-Casado et al. (2019)) | 512 | FP | FP | FP | 98.7 | 1046 |
| QORNN (Foucault et al. (2024)) | 170 | 8 | 8 | 12 | 96.2 | 35.0 |
| QORNN (Foucault et al. (2024)) | 170 | 6 | 6 | 12 | 94.74 | 27.9 |
| | | | 2 | FP | 92.65 | 3.48 |
| | | | 4 | FP | 96.63 | 4.85 |
| HadamRNN (ours) | 512 | 1 | 4 | 12 | 96.34 | 3.58 |
| | | | 6 | FP | 96.9 | 6.23 |
| **IMDB** ($T = 500$, $d_{in} = 512$, $d_{out} = 1$, accuracy) | | | | | | |
| LSTM (Alom et al. (2018)) | 128 | | 1 | FP | 76.25 | 40.50 |
| LSTM (Alom et al. (2018)) | 128 | | 2 | FP | 79.64 | 80.5 |
| LSTM (He et al. (2016)) | 512 | 0 | 2 | 2 | 88.12 | 514 |
| LSTM (He et al. (2016)) | 512 | | 4 | 4 | 88.48 | 1026 |
| ORNN | 128 | FP | FP | FP | 84.02 | 320.5 |
| | | | ternary | FP | 81.18 | 16.55 |
| | 128 | | 2 | FP | 85.34 | 16.55 |
| HadamRNN (ours) | | | 2 | 12 | 85.18 | 16.24 |
| | 512 | 1 | 4 | FP | 87.43 | 130.32 |
| | | | 4 | 12 | 87.13 | 129.06 |

For a given size $d_h$, reducing $q$ improves performance. On the contrary, increasing $q$ improves computational complexity. For instance, Block-HadamRNN with $q = 8$ consistently solves the copy task with only 2048 fixed-point precision additions, compared to the $16,384$ ones required by the HadamRNN. In addition, for the copy-task and sMNIST, Block-HadamRNN with $q = 2$ has a negligible drop of accuracy compared to HadamRNN, while calculating a product with its recurrent weight matrix requires only half the number of fixed-point precision additions. This, along with the plots in Appendix E.3 based on the results from Table 3, demonstrates that Block-HadamRNN facilitates exploration of the trade-off between computational complexity and performance, offering flexible control over both performance and resource utilization.

Table 3: Comparison of the performances of Block-HadamRNN (with paramer $q$) and HadamRNN ($q = 1$) on several benchmarks. The last column reports the computational complexity of the recurrent operation $W h_t$, measured in fixed-point precision additions (see Table 1). The quantization bitwith of matrices $U$ and $V$ is 4 and the activations are not quantized.

| Model | $d_h$ | W bitwidth | parameter $q$ in (4) | performance | computational complexity |
|---|---|---|---|---|---|
| **Copy task** ($T = 1000$, $d_{in} = 10$, $d_{out} = 9$, cross-ent, baseline = 0.021) | | | | | |
| Block-HadamRNN | 128 | ternary | 32 | 1.6e-3 | 512 |
| | | | 8 | 6.6e-5 | 2048 |
| | | | 2 | 2.0e-6 | 8192 |
| HadamRNN | 128 | 1 | 1 | 1.6e-7 | 16,384 |
| **Permuted MNIST** ($T = 784$, $d_{in} = 1$, $d_{out} = 10$, accuracy) | | | | | |
| Block-HadamRNN | 512 | ternary | 128 | 60.74 | 2048 |
| | | | 32 | 91.42 | 8192 |
| | | | 8 | 91.45 | 32,768 |
| | | | 2 | 93.12 | 131,072 |
| HadamRNN | 512 | 1 | 1 | 94.88 | 262,144 |
| **Sequential MNIST** ($T = 784$, $d_{in} = 1$, $d_{out} = 10$, accuracy) | | | | | |
| Block-HadamRNN | 512 | ternary | 128 | 27.45 | 2048 |
| | | | 32 | 80.60 | 8192 |
| | | | 8 | 92.49 | 32,768 |
| | | | 2 | 96.47 | 131,072 |
| HadamRNN | 512 | 1 | 1 | 96.63 | 262,144 |
| **IMDB** ($T = 500$, $d_{in} = 512$, $d_{out} = 1$, accuracy) | | | | | |
| Block-HadamRNN | 512 | ternary | 128 | 81.83 | 2048 |
| | | | 32 | 84.27 | 8192 |
| | | | 8 | 85.70 | 32,768 |
| | | | 2 | 86.30 | 131,072 |
| HadamRNN | 512 | 1 | 1 | 87.43 | 262,144 |

## 5 CONCLUSION

Drawing on Hadamard matrix theory, this article presents a method for parameterizing a subset of binary and sparse ternary orthogonal matrices. We demonstrate that the parameters of such matrices can be learned using standard methods like the straight-through estimator (STE), and empirically validate that this subset is sufficiently expressive to solve standard RNN benchmarks. This work is the first to construct efficient orthogonal RNNs with binary and sparse ternary recurrent weight matrices. This was recognized as a challenging problem by Ott et al. (2016) and has not been addressed since. Experimental results show that the proposed HadamRNN matches the performance of floating-point ORNNs while reducing the model size by up to 290-fold. Notably, it is the first binary recurrent weight model capable of learning the copy task with more than 1,000 timesteps. With the proposed sparse-ternary models, Block-HadamRNN, we offer ways to fine-tune the balance between performance and computational efficiency. Future work could explore the following directions: (1) binarizing or ternarizing Structured State Space Models to address tasks with even longer-range dependencies, such as those in the Long Range Arena benchmark (Tay et al., 2020); (2) deploying HadamRNNs on edge devices; and (3) applying binary orthogonal matrices to other domains, including time series forecasting (Wu et al., 2021; Zhou et al., 2021; 2022), neural network robustness (Cisse et al., 2017; Anil et al., 2019), Normalizing Flows (Kingma and Dhariwal, 2018), and Wasserstein distance estimation (Brock et al., 2018).

ACKNOWLEDGMENTS AND DISCLOSURE OF FUNDING

This work has benefited from the AI Interdisciplinary Institute ANITI, which is funded by the French "Investing for the Future – PIA3" program under the Grant agreement ANR-19-P3IA-0004. The authors gratefully acknowledge the support of the DEEL project.[3] A. Foucault was supported by 'Région Occitanie, France', which provided a PhD grant. Part of this work was performed using HPC resources from CALMIP (Grant 2024-P22034).

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

## A   Bibliography on ORNNs

This appendix provides a detailed bibliography on unitary and orthogonal recurrent neural networks.

Unitary Recurrent Neural Networks (URNNs) were introduced in Arjovsky et al. (2016) to capture long-term dependencies more effectively than LSTMs. Several methods have been developed to parameterize recurrent weight matrices in URNNs and orthogonal RNNs. These include using the Cayley transform (Wisdom et al., 2016; Helfrich et al., 2018), Givens rotations (Jing et al., 2017), Householder reflections (Mhammedi et al., 2017), Kronecker matrices (Jose et al., 2018), soft-orthogonality (Vorontsov et al., 2017), the Singular Value Decomposition (SVD) (Zhang et al., 2018), the exponential map (Lezcano-Casado et al., 2019), and Riemannian optimization strategies (Kiani et al., 2022). Each method aims to improve model expressivity, efficiency or reducing complexity.

The only known attempt to quantize ORNN weights is detailed in Foucault et al. (2024). This method enables the learning of challenging tasks, such as the copy task for 1000 timesteps, using 5 bits for the weights and 12 bits for the activations.

## B   Proofs

We begin this section by reviewing the definition and a proposition on the Kronecker product. Then for completeness, in Appendix B.2, we provide the proof of Proposition 3.2 and, in Appendix B.3, we provide the proof of Proposition 3.3.

### B.1   Reminders on the Kronecker product

**Definition B.1.** Let $p, q, r, s \in \mathbb{N}^*$. Let $A = \left(a_{ij}\right)_{ij} \in \mathbb{R}^{p \times q}$ and $B \in \mathbb{R}^{r \times s}$. The Kronecker product of $A$ by $B$, denoted $A \otimes B$, is the matrix of size $pr \times qs$ given by

$$A \otimes B = \begin{pmatrix} a_{11} B & \dots & a_{1q} B \\ \vdots & \ddots & \vdots \\ a_{p1} B & \dots & a_{pq} B \end{pmatrix}.$$

The following proposition states a well-known result concerning the Kronecker product.

**Proposition B.2.** *Let $p, q, r, s \in \mathbb{N}^*$. Let $A \in \mathbb{R}^{p \times q}$ and $B \in \mathbb{R}^{r \times s}$. If the lines of $A$ are pairwise orthogonal and the lines of $B$ are pairwise orthogonal, then the lines of $A \otimes B$ are pairwise orthogonal.*

We provide the proof for completeness.

*Proof.* Let $A \in \mathbb{R}^{p \times q}$ and $B \in \mathbb{R}^{r \times s}$ be two matrices. We denote $A_i \in \mathbb{R}^q$ (resp $B_i \in \mathbb{R}^s$) the $i^{\text{th}}$ line of $A$ (resp $B$). Assume that for all $(i, j) \in [\![1, p]\!]^2$ satisfying $i \neq j$, $A_i A_j' = 0$. Assume also that for all $(m, n) \in [\![1, r]\!]^2$ satisfying $m \neq n$, $B_m B_n' = 0$.

The hypotheses imply that there is $\alpha \in \mathbb{R}^p$ and $\beta \in \mathbb{R}^r$ such that

$$AA' = \text{diag}(\alpha) \qquad \text{and} \qquad BB' = \text{diag}(\beta).$$

Denoting

$$\begin{aligned} C & = A \otimes B \\ & = \begin{pmatrix} a_{11} B & \dots & a_{1q} B \\ \vdots & \ddots & \vdots \\ a_{p1} B & \dots & a_{pq} B \end{pmatrix}. \end{aligned}$$

For any $(i, j) \in [\![1, p]\!]$, using block matrix multiplication, the block of size $r \times r$ at position $(i, j)$ of $CC'$ is

$$\sum_{k=1}^{q} (a_{i,k} B)(a_{j,k} B)' = \sum_{k=1}^{q} a_{i,k} a_{j,k} \, \text{diag}(\beta) = \text{diag}(\alpha)_{i,j} \, \text{diag}(\beta) = \begin{cases} 0 & \text{if } i \neq j \\ \alpha_i \, \text{diag}(\beta) & \text{if } i = j \end{cases}.$$

Therefore $CC' = \text{diag}(\alpha \otimes \beta)$ and the lines of $A \otimes B$ are pairwise orthogonal.

□

## B.2 PROOF OF PROPOSITION 3.2

We proceed by induction.

- **Initialization:** Consider $k = 1$. Using the definition of $\mathbf{S}_2$, we have

$$\mathbf{S}_2\,\mathbf{S}_2' = \begin{pmatrix} 1 & 1 \\ 1 & -1 \end{pmatrix} \begin{pmatrix} 1 & 1 \\ 1 & -1 \end{pmatrix} = 2I_2.$$

  Therefore the lines of $\mathbf{S}_2$ are pairwise orthogonal. Since $\mathbf{S}_2$ is square and its components are in $\{-1, +1\}$, $\mathbf{S}_2$ is a Hadamard matrix.

- **Heredity:** Consider $k \geq 1$ and assume that $\mathbf{S}_{2^k}$ is a Hadamard matrix. We want to prove that $\mathbf{S}_{2^{k+1}} = \begin{pmatrix} \mathbf{S}_{2^k} & \mathbf{S}_{2^k} \\ \mathbf{S}_{2^k} & -\mathbf{S}_{2^k} \end{pmatrix}$ is a Hadamard matrix.

  Notice first that since $\mathbf{S}_{2^k}$ is a Hadamard matrix, $\mathbf{S}_{2^{k+1}}$ is square and its components are in $\{-1, +1\}$. Using the definition of the Kronecker product, we also have

$$\mathbf{S}_{2^{k+1}} = \begin{pmatrix} 1 & 1 \\ 1 & -1 \end{pmatrix} \otimes \mathbf{S}_{2^k} = \mathbf{S}_2 \otimes \mathbf{S}_{2^k}.$$

  Since the lines of $\mathbf{S}_2$ are pairwise orthogonal and the lines of $\mathbf{S}_{2^k}$ are pairwise orthogonal, we can apply Proposition B.2 and conclude that the lines of $\mathbf{S}_{2^{k+1}}$ are also pairwise orthogonal. We conclude that $\mathbf{S}_{2^{k+1}}$ is a Hadamard matrix.

This concludes the proof by induction.

## B.3 PROOF OF PROPOSITION 3.3

Let $n \in \mathbb{N}^*$ and $H \in \mathcal{H}_n$. We first show that $\phi_H$ is injective.

Let $u, u' \in \{-1, 1\}^n$ such that $u \neq u'$. Let $i \in \{1, \dots, n\}$ be such that $u_i \neq u_i'$, that is, since $u_i$ and $u_i'$ are both in $\{-1, 1\}$, $u_i = -u_i'$. Denoting, for all matrix $A$, the $i$-th row of $A$ by $A_i$, we obtain

$$\phi_H(u)_i = u_i H_i = -u_i' H_i = -\phi_H(u')_i$$

Since all the components of $\phi_H(u)_i$ are in $\{-1, 1\}$, $\phi_H(u)_i \neq 0$ and finally $\phi_H(u)_i \neq \phi_H(u')_i$.

As a conclusion, for any $u, u' \in \{-1, 1\}^n$ such that $u \neq u'$, $\phi_H(u) \neq \phi_H(u')$. The mapping $\phi_H$ is injective.

We now show that $\phi_H(u)$ is a Hadamard matrix. Notice first that $\phi_H(u)$ is square and that all its components are $\{-1, 1\}$. We still need to show that any two distinct rows of $\phi_H(u)$ are orthogonal. Let $i, j \in \{1, \dots, n\}$ with $i \neq j$. Reminding that $\phi_H(u)_i$ is $i$-th line of $\phi_H(u)$, we have

$$\phi_H(u)_i \phi_H(u)_j' = u_i u_j H_i H_j' = 0.$$

Finally, $\phi_H(u)$ is a Hadamard matrix.

This concludes the proof.

## B.4 DETAILED PROOF OF THE ORTHOGONALITY OF THE BINARY AND SPARSE TERNARY WEIGHTS

The proof that the binary matrix defined by

$$W(u) = \frac{1}{\sqrt{d_h}}\,\mathrm{diag}(u)\,\mathbf{S}_{2^k}\ \in \mathbb{R}^{d_h \times d_h},$$

is orthogonal when $d_h = 4$ is in Appendix C.2. The general proof is similar to the proof that the sparse ternary matrix defined below is orthogonal. We only detail the latter proof.

Let us prove that the sparse ternary weights defined by

$$W(u) = \frac{1}{\sqrt{2^k}} \operatorname{diag}(u) \left(I_q \otimes \mathbf{S}_{2^k}\right) \ \in \mathbb{R}^{d_h \times d_h}, \tag{5}$$

are orthogonal for all $u \in \{-1, 1\}^{d_h}$, and $d_h = q2^k$.

To do so, we consider $u \in \{-1, 1\}^{d_h}$. We first remark that the lines of $I_q$ are pairwise orthogonal. Because $\mathbf{S}_{2^k}$ is a Hadamard matrix, the lines of $\mathbf{S}_{2^k}$ are also pairwise orthogonal. Applying Proposition B.2, we conclude that the lines of $I_q \otimes \mathbf{S}_{2^k}$ are pairwise orthogonal. Therefore, the lines of $W(u)$ are also pairwise orthogonal and the matrix $W(u)W(u)'$ is diagonal. Let us consider $i \in [\![1, d_h]\!]$, we write $i = (m-1)2^k + n$, where $m \in [\![1, q]\!]$ and $n \in [\![1, 2^k]\!]$. Reminding that $W(u)_i$ is the $i$-th line of $W(u)$, and $(\mathbf{S}_{2^k})_n$ is the $n$-th line of $\mathbf{S}_{2^k}$, we have

$$\begin{aligned}
\left(W(u)W(u)'\right)_{i,i} = W(u)_i (W(u)_i)' &= \left(\frac{1}{\sqrt{2^k}} u_i (\mathbf{S}_{2^k})_n\right)\left(\frac{1}{\sqrt{2^k}} u_i (\mathbf{S}_{2^k})_n\right)' \\
&= \frac{1}{2^k} u_i^2 \sum_{j=1}^{2^k} (\mathbf{S}_{2^k})_{n,j}^2 \\
&= 1
\end{aligned}$$

because $u_i \in \{-1, 1\}$ and all the components of $\mathbf{S}_{2^k}$ are in $\{-1, 1\}$.

Finally, we conclude that $W(u)W(u)' = I_{d_h}$. Because the matrix $W(u)$ is square, we also have $W(u)'W(u) = I_{d_h}$ and the matrix $W(u)$ is orthogonal.

This concludes the proof the sparse ternary matrix defined by (5) is orthogonal.

## C  THE STRAIGHT-THROUGH ESTIMATOR AND AN EXAMPLE

We present the Straight-through Estimator (STE) in Appendix C.1 and provide in Appendix C.2 a detailed example of a recurrent weight matrix for the HadamRNN defined in Section 3.3, specifically when $d_h = 4$.

### C.1  THE STRAIGHT-THROUGH ESTIMATOR

In this section, we discuss the Straight-through Estimator, introduced in Hinton (2012); Bengio et al. (2013); Courbariaux et al. (2015), a standard method for optimizing quantized neural network weights, in the context of HadamRNN and Block-HadamRNN.

For simplicity, we omit the optimization of $U$, $V$, $b_i$ and $b_o$ in the following description, focusing on the recurrent matrix.

We consider a matrix $W \in \mathbb{R}^{d_h \times d_h}$. For the HadamRNNs defined in Section 3.3, we use the constant orthogonal matrix $W = \frac{1}{\sqrt{d_h}} \mathbf{S}_{2^k}$, while for the Block-HadamRNNs defined in Section 3.4, we take $W = \frac{1}{\sqrt{2^k}}\left(I_q \otimes \mathbf{S}_{2^k}\right)$. As described in these sections, the only trainable parameter for the recurrent matrix is the binary vector $u$ which defines the recurrent weight matrix $\operatorname{diag}(u)W$.

We will describe the STE method for optimizing $u \in \{-1, 1\}^{d_h}$. Therefore, we only consider a learning objective $L : \mathbb{R}^{d_h} \longrightarrow \mathbb{R}$ and the optimization problem

$$\operatorname{argmin}_{u \in \{-1,1\}^{d_h}} L(u). \tag{6}$$

To optimize $u$, we define the quantization operator $H : \mathbb{R}^{d_h} \longrightarrow \{-1, 1\}^{d_h}$, where, for all $\tilde{u} \in \mathbb{R}^{d_h}$, the vector $H(\tilde{u}) \in \{-1, 1\}^{d_h}$ is given by

$$H(\tilde{u})_i = \begin{cases} +1 & \text{if } \tilde{u}_i \geq 0 \\ -1 & \text{otherwise} \end{cases} \quad , \text{for all } i \in \{1, \dots, d_h\}.$$

The operator $H$ is surjective, since for all $u \in \{-1, 1\}^{d_h}$, $H(u) = u$. Therefore, (6) is equivalent to minimizing $L \circ H$ over $\mathbb{R}^{d_h}$. To minimize $L \circ H$ the STE applies a modified gradient descent algorithm. The modification is described below.

The operator $H$ is piecewise constant and its gradient at $\tilde{u}$, denoted $\left. \frac{\partial H}{\partial \tilde{u}} \right|_{\tilde{u}}$, is either undefined or 0. This issue is standard in quantization-aware training, which aims to minimize the objective $L(H(\tilde{u}))$ with respect to $\tilde{u}$. Backpropagating the gradient using the chain rule

$$\left. \frac{\partial L \circ H}{\partial \tilde{u}} \right|_{\tilde{u}} = \left. \frac{\partial L}{\partial u} \right|_{H(\tilde{u})} \left. \frac{\partial H}{\partial \tilde{u}} \right|_{\tilde{u}}$$

is either not possible or results in a null gradient in this context.

To address this issue, backpropagation through the quantization operator $H$ is performed using STE Hinton (2012); Bengio et al. (2013); Courbariaux et al. (2015). The latter approximates the gradient using

$$\left. \frac{\partial L \circ H}{\partial \tilde{u}} \right|_{\tilde{u}} \approx \left. \frac{\partial L}{\partial u} \right|_{H(\tilde{u})},$$

as if $\left. \frac{\partial H}{\partial \tilde{u}} \right|_{\tilde{u}} = I_{d_h}$.

## C.2 EXAMPLE OF A RECURRENT WEIGHT MATRICE

For HadamRNN, with $d_h = 4$, using (3) and Proposition 3.2, the recurrent weight matrices are defined for $u \in \{-1, +1\}^4$ by

$$W(u) = \frac{1}{2} \begin{pmatrix} u_1 & 0 & 0 & 0 \\ 0 & u_2 & 0 & 0 \\ 0 & 0 & u_3 & 0 \\ 0 & 0 & 0 & u_4 \end{pmatrix} \begin{pmatrix} 1 & 1 & 1 & 1 \\ 1 & -1 & 1 & -1 \\ 1 & 1 & -1 & -1 \\ 1 & -1 & -1 & 1 \end{pmatrix}.$$

The components of $u$ are optimized using the STE, as described in Appendix C.1.

As mentioned in Section 3.3 and demonstrated in Appendix B.4 for the general case, it can be verified that, for all $u \in \{-1, +1\}^4$, $W(u)$ is orthogonal, i.e. $W(u)W(u)' = I_{d_h}$. Indeed, $W(u)W(u)'$ equals

$$\frac{1}{4} \begin{pmatrix} u_1 & 0 & 0 & 0 \\ 0 & u_2 & 0 & 0 \\ 0 & 0 & u_3 & 0 \\ 0 & 0 & 0 & u_4 \end{pmatrix} \begin{pmatrix} 1 & 1 & 1 & 1 \\ 1 & -1 & 1 & -1 \\ 1 & 1 & -1 & -1 \\ 1 & -1 & -1 & 1 \end{pmatrix} \begin{pmatrix} 1 & 1 & 1 & 1 \\ 1 & -1 & 1 & -1 \\ 1 & 1 & -1 & -1 \\ 1 & -1 & -1 & 1 \end{pmatrix} \begin{pmatrix} u_1 & 0 & 0 & 0 \\ 0 & u_2 & 0 & 0 \\ 0 & 0 & u_3 & 0 \\ 0 & 0 & 0 & u_4 \end{pmatrix}$$

$$= \frac{1}{4} \begin{pmatrix} u_1 & 0 & 0 & 0 \\ 0 & u_2 & 0 & 0 \\ 0 & 0 & u_3 & 0 \\ 0 & 0 & 0 & u_4 \end{pmatrix} \begin{pmatrix} 4 & 0 & 0 & 0 \\ 0 & 4 & 0 & 0 \\ 0 & 0 & 4 & 0 \\ 0 & 0 & 0 & 4 \end{pmatrix} \begin{pmatrix} u_1 & 0 & 0 & 0 \\ 0 & u_2 & 0 & 0 \\ 0 & 0 & u_3 & 0 \\ 0 & 0 & 0 & u_4 \end{pmatrix}$$

$$= \begin{pmatrix} u_1^2 & 0 & 0 & 0 \\ 0 & u_2^2 & 0 & 0 \\ 0 & 0 & u_3^2 & 0 \\ 0 & 0 & 0 & u_4^2 \end{pmatrix} = I_{d_h}.$$

## D EXPERIMENTS DETAILS

### D.1 COPY TASK

We generated $512K$ samples for the training set, and $2K$ samples for both validation and test. HadamRNN and Block-HadamRNN were trained using the Adam optimizer Kingma and Ba (2015). We used a batch size of 128 samples. The learning rate is initialized to $1e-4$ is decayed exponentially by applying a factor 0.98 after each epoch. 10 epochs were used for training.

## D.2 PERMUTED / SEQUENTIAL MNIST

We used $50K$ samples for training, $10K$ samples for validation and $10K$ samples for testing. Hadam-RNN and Block-HadamRNN were trained using the Adam optimizer Kingma and Ba (2015). We used a batch size of 64 samples. The learning rate is initialized to $1e-3$ is decayed exponentially by applying a factor 0.98 after each epoch. 200 epochs were used for training.

## D.3 IMDB DATASET

The IMDB dataset contains $50,000$ samples. Among these, $25,000$ samples are used for training, and the remaining $25,000$ are equally divided between validation and testing.

We used a batch size of 100 samples. The learning rate is initialized to $5e-4$ is decayed exponentially by applying a factor 0.99 after each epoch. 30 epochs were used for training.

# E COMPLEMENTS ON THE EXPERIMENTS

## E.1 TRAINING TIME

For each benchmark of Section 4 and Appendix F.1, we provide the training time for the HadamRNN in Table 4. As the training times for the Block-HadamRNN are comparable, they are not explicitly reported here. Experiments where done on a NVIDIA GeForce RTX 3080 GPU. For comparison, FastGRNN Kusupati et al. (2018) requires 16.97 hours of training time on the MNIST dataset. The training times for the HAR-2 and DSA-19 datasets, along with a comparison to those of FastGRNN, are presented in Table 7.

Table 4: Training times, in hours

| dataset | Copy task | pMNIST / sMNIST | IMDB | SST-2 | QQP |
|---|---|---|---|---|---|
| # epochs | 10 | 200 | 30 | 60 | 50 |
| training time (hr) | 13 | 13 | 0.5 | 0.36 | 7.5 |

## E.2 TRAINING STABILITY

In this section, we examine the variability of the trained HadamRNN model based on different random initializations and randomness in the stochastic algorithm (Adam). The experiments were conducted using the IMDB dataset, with similar trends observed across the other benchmarks, supporting the consistency of these conclusions.

Each trainable matrix or vector is initialized using the Glorot initialization method Glorot and Bengio (2010), which relies on a random seed. For a fixed combination of bitwidth $p$ (applied to the matrices $U$ and $V$) and initial learning rate, we train 5 different models using 5 different random seeds. The average performance and standard deviation of the models are reported in Table 5.

As expected, the initial learning rate significantly influences the performance of the trained model. It appears that performance improves as the initial learning rate increases when $p$ decreases. In all cases, the optimal learning rate spans approximately one order of magnitude, suggesting that the search for the optimal initial learning rate can be conducted on an exponential scale.

A bitwidth of $p = 2$ results in relatively high variability in the performance of the trained model, whereas bitwidths of $p = 4$ and $p = 6$ exhibit negligible variance. As anticipated, smaller bitwidths on U and V matrices make the training process more sensitive to the random initialization of the parameters.

## E.3 VISUALIZATION OF MODEL COMPARISONS

We provide two visual comparisons of HadamRNN, Block-HadamRNN, QORNN, LSTM and ORNN based on different criteria, using data from Table 2 and Table 3.

Table 5: HadamRNN's training variability over 5 experiments on the IMDB dataset, for each combination of bitwidth $p$, for the matrices $U$ and $V$, and initial learning rate.

| U & V bitwidth | Initial learning rate | Average performance | Standard deviation |
|---|---|---|---|
| 2 | 1.e-2 | 83.28 | 1.00 |
| | 5.e-3 | 83.83 | 1.36 |
| | 1.e-3 | 80.84 | 5.09 |
| | 5.e-4 | 80.75 | 8.05 |
| | 1.e-4 | 74.61 | 2.69 |
| 4 | 1.e-2 | 83.76 | 0.70 |
| | 5.e-3 | 83.73 | 0.40 |
| | 1.e-3 | 85.83 | 0.40 |
| | 5.e-4 | 85.85 | 0.39 |
| | 1.e-4 | 83.05 | 0.49 |
| 6 | 1.e-2 | 83.18 | 1.88 |
| | 5.e-3 | 83.61 | 0.53 |
| | 1.e-3 | 85.73 | 0.63 |
| | 5.e-4 | 86.09 | 0.35 |
| | 1.e-4 | 83.47 | 0.50 |

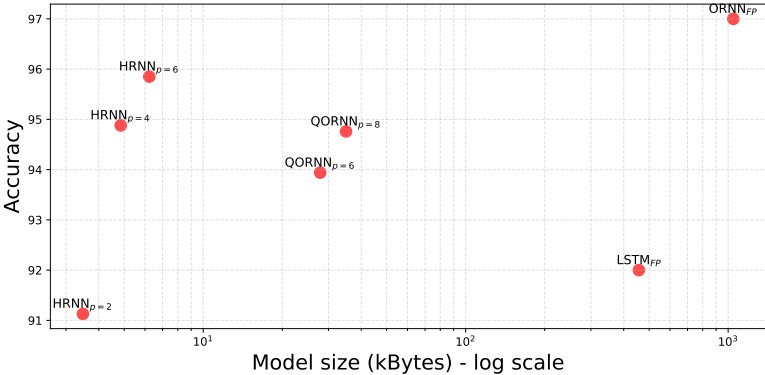

Figure 1: Position of each model in the (size, performance) plane, on pMNIST. The most effective models are located in the upper-left corner of Figure 1. The parameter $p$ corresponds to the bitwidth of the quantized matrices $U$ and $V$, as introduced in Section 3.5. *FP* stands for *full-precision*.

- In Figure 1, we plot each model in the (size, performance) plane, where size is measured in kilobytes (kB) and performance corresponds to the accuracy of the trained model on pMNIST. The most effective models are located in the upper-left corner of Figure 1. The plot highlights the efficiency of HadamRNN.

- In Figure 2, we plot the HadamRNN and Block-HadamRNN models for different values of $q$ in the (complexity, performance) plane. Complexity is measured by the number of fixed-point precision additions (see Table 1), while performance corresponds to the cross-entropy of the trained model on the Copy task. The most effective models appear in the lower-left corner of Figure 2.

  This visualization highlights that, for the Copy task, $q$ can be tuned to balance the trade-off between complexity and performance.

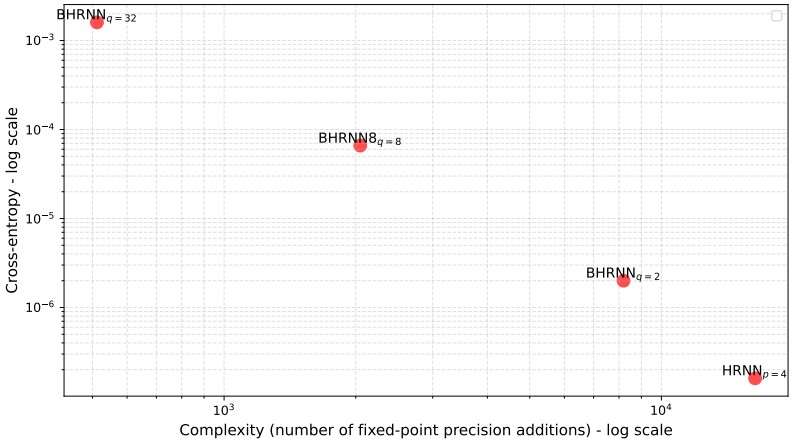

Figure 2: Position of HadamRNN and Block-HadamRNN models for different values of $q$ in the (complexity, performance) plane. The most effective models appear in the lower-left corner of Figure 2. The bitwidth of the quantized matrices $U$ and $V$ is set to $p = 4$.

## F MORE BENCHMARKS

In Appendix F.1, we provide a comparison of HadamRNN and quantized versions of BERT on the SST-2 and QQP benchmarks from GLUE (Wang et al., 2019). In Appendix F.2, we provide results on IoT task benchmarks: HAR-2 and DSA-19 as described in Kusupati et al. (2018).

### F.1 GLUE BENCHMARK AND COMPARISON WITH TRANSFORMERS

GLUE is a well-established benchmark for NLP systems, comprising various sentence-level language understanding tasks. The most effective solutions for these tasks are large models trained on extensive datasets, typically large pretrained language models (LLMs) followed by fine-tuning with a task-specific head.

Although HadamRNN and Block-HadamRNN are not designed to operate within a large language model (LLM) framework, and no ORNN results are reported on the benchmark leaderboard, we evaluate the performance of HadamRNN and Block-HadamRNN trained directly on the training data of a single task. This allows us to position these networks relative to transformer-based models.

We selected two tasks: SST-2, a sentiment analysis task closely related to the objective of the IMDB dataset, and QQP (Quora Question Pairs), which evaluates whether two questions are semantically equivalent. For both tasks, as is standard practice, we used a tokenizer to map sentences into sequences of indices, padded to a fixed length of 128 timesteps. For sentence pairs, as recommended (and done in the reference code), we concatenated both sentences with two separator tokens in between.

Each task was trained using the respective training dataset without data augmentation. The network achieving the best score on the validation dataset was retained, and its predictions were submitted to the GLUE Benchmark website. The performance metrics provided by the website are reported in Table 6.

Since HadamRNN and Block-HadamRNN are trained exclusively on the training data for a single task, larger networks tend to overfit. Notably, we were unable to train and generalize effectively on validation data using pre-trained embeddings, such as the first layer of BERT (with a size of 768). Instead, we used a trainable embedding layer of size 256.

For the SST-2 task, the best results were achieved with an HadamRNN network having a hidden size of 128, resulting in a total model size of approximately 8 kB and an accuracy of 81.9%. In comparison, the smallest BiBERT model, distilled from a BERT pre-trained on a large dataset and with a network size 550 times larger, achieves 85.4%.

Table 6: Performance comparison of HadamRNN, Block-HadamRNN, and state-of-the-art models on two GLUE benchmark tasks: SST-2 and QQP. The last column indicates the model size in kilobytes (kB).

| Model | Performance | Size kBytes |
|---|---|---|
| **SST-2 task** ($T = 128$, $d_{out} = 1$, accuracy) | | |
| Q8BERT (Zafrir et al., 2019) | 92.24 | 110 000 |
| Q-BERT (Shen et al., 2020) | 92.08 | 48 100 |
| TernaryBERT (Zhang et al., 2020) | 92.8 | 18 000 |
| MobileBERT (Sun et al., 2020) | 91.6 | 15 100 |
| BinaryBERT (Bai et al., 2021) | 92.3 | 13 400 |
| BiT (Liu et al., 2022) | 89.9 | 13 400 |
| BiBERT (Qin et al. (2022)) | 88.7 | 13 400 |
| BiBERT (Qin et al., 2022) | 87.9 | 6 800 |
| BiBERT (Qin et al., 2022) | 85.4 | 4 400 |
| HadamRNN (ours, $d_i = 256$, $d_h = 128$) | 81.9 | 8 |
| **QQP task** ($T = 128$, $d_{out} = 1$, accuracy) | | |
| Q8BERT (Zafrir et al., 2019) | 87.96 | 110 000 |
| TernaryBERT (Zhang et al., 2020) | 88.8 | 18 000 |
| BinaryBERT (Bai et al., 2021) | 88.9 | 13 400 |
| BiT (Liu et al., 2022) | 85.4 | 13 400 |
| BiBERT (Qin et al., 2022) | 84.8 | 13 400 |
| BiBERT (Qin et al., 2022) | 83.3 | 6 800 |
| BiBERT (Qin et al., 2022) | 78.2 | 4 400 |
| Block-HadamRNN (ours, $d_i = 256$, $d_h = 512$, $q = 128$) | 82.1 | 33 |

For the QQP task, using a Block-HadamRNN network with a hidden size of 512 and a block size of 128, we obtained an accuracy of 82.1%. This result surpasses the smallest BiBERT, despite our model being more than 130 times smaller.

## F.2  INTERNET OF THINGS (IoT) BENCHMARKS

HadamRNN and Block-HadamRNN are particularly well-suited for modeling long-term dependencies such as the Copy task, thanks to their orthogonal recurrent weight matrix. To provide a comprehensive evaluation, we also tested these models on two short-term memory IoT benchmarks (< 150 timesteps), previously addressed by FastGRNN Kusupati et al. (2018). The results of these experiments are summarized in Table 7. For both benchmarks, the hyperparameter $q$ in Block-HadamRNN (detailed in Section 3.4) was optimized using grid search.

**Human Activity Recognition (HAR-2)**   The Human Activity Recognition (HAR) dataset consists of human motion data captured using an accelerometer and gyroscope embedded in a Samsung Galaxy S II smartphone. Data is recorded at a fixed frequency of 50 Hz, with each sequence spanning 128 timesteps. Each sequence is labeled with one of the following six activity classes: *Sitting*, *Laying*, *Walking_Upstairs*, *Standing*, *Walking*, and *Walking_Downstairs*. Following the approach in Kusupati et al. (2018), these six classes are grouped into two categories: {Sitting, Laying, Walking_Upstairs} and {Standing, Walking, Walking_Downstairs}. The objective is to classify each sequence into the correct category. Both the training and test sets have been preprocessed to ensure zero mean and unit variance, ensuring consistency across the data.

**Daily and Sports Activity (DSA-19)**   This dataset comprises motion sensor data collected from accelerometers, gyroscopes, and magnetometers, capturing a range of daily and sports-related human activities. Measurements are sampled at a fixed frequency of 25 Hz and segmented into sequences of 125 timesteps. A total of 19 distinct activities were performed by the participants, with each sequence labeled according to its corresponding activity type. The objective is to accurately predict the activity

Table 7: Comparison of performances of the HadamRNN, the Block-HadamRNN and FastGRNN Kusupati et al. (2018) on two IoT benchmarks (HAR-2 and DSA-19). Last column reports the model size in kBytes.

| Model | $d_h$ | train time (min) | W bitwidth | U & V bitwidth | performance | size kBytes |
|---|---|---|---|---|---|---|
| **HAR-2** ($T = 128$, $d_{in} = 9$, $d_{out} = 1$, accuracy) | | | | | | |
| FastGRNN (Kusupati et al. (2018)) | 80 | 4.8 | 8 | 8 | 95.59 | 3.00 |
| Block-HadamRNN ($q = 2$) | 1024 | 12.1 | | 4 | 94.81 | 9.13 |
| | | 12.5 | 1 | 6 | 95.76 | 11.63 |
| | | 10.2 | | 8 | 95.36 | 14.13 |
| | 512 | 12.0 | | 4 | 94.87 | 4.57 |
| | | 12.1 | 1 | 6 | 94.56 | 5.82 |
| | | 12.2 | | 8 | 95.64 | 7.07 |
| HadamRNN | 512 | 4.4 | | 4 | 83.40 | 4.57 |
| | | 5.2 | 1 | 6 | 84.45 | 5.82 |
| | | 4.2 | | 8 | 84.11 | 7.07 |
| **DSA-19** ($T = 125$, $d_{in} = 45$, $d_{out} = 19$, accuracy) | | | | | | |
| FastGRNN (Kusupati et al. (2018)) | 80 | 2.2 | 8 | 8 | 85.67 | 22.00 |
| Block-HadamRNN ($q = 16$) | 256 | 5.1 | | 4 | 76.51 | 9.11 |
| | | 6.1 | 1 | 6 | 78.30 | 13.11 |
| | | 3.0 | | 8 | 75.5 | 17.11 |
| | 512 | 5.2 | | 4 | 81.32 | 18.14 |
| | | 6.2 | 1 | 6 | 85.26 | 26.14 |
| | | 3.2 | | 8 | 81.86 | 34.14 |
| HadamRNN | 512 | 5.2 | | 4 | 77.93 | 18.14 |
| | | 3.4 | 1 | 6 | 76.01 | 26.14 |
| | | 3.3 | | 8 | 79.12 | 34.14 |

associated with each sequence. Both the training and test sets have been preprocessed to achieve zero mean and unit variance, ensuring consistency across the data.

Our results show that an optimized Block-HadamRNN delivers performance comparable to Fast-GRNN while maintaining a similarly compact model size. For instance, on the DSA-19 benchmark, the best Block-HadamRNN model achieves performance within approximately 0.4% of FastGRNN, with a comparable size. Notably, on the HAR-2 benchmark, the Block-HadamRNN outperforms FastGRNN in performance with only a marginal size increase of a few kilobytes. These findings suggest that HadamRNN and Block-HadamRNN are also well-suited for tasks where long-term memory capabilities are not a critical requirement.

## G    LINEAR VERSUS RELU RECURRENT UNITS

In this appendix, we argue that 'linear-ORNNs', as considered in Section 3.1, are better suited than 'ReLU-ORNNs' for tasks that require strong memorization. An ablation study supporting this fact is in Appendix G.1. The theoretical arguments are given in Appendix G.2.1 and the experimental arguments are in Appendix G.2.2.

Throughout the appendix, we consider $h_0 = 0$, a matrix $U \in \mathbb{R}^{d_h \times d_{in}}$ and a matrix $V \in \mathbb{R}^{d_{out} \times d_h}$, $b_i \in \mathbb{R}^{d_h}$, $b_o \in \mathbb{R}^{d_{out}}$, an orthogonal recurrent weight matrix $W \in \mathbb{R}^{d_h \times d_h}$ and the ReLU activation function $\sigma$. We also consider inputs $x_1, \ldots, x_T \in \mathbb{R}^{d_{in}}$.

- We call *linear-ORNNs*, those computing the sequence of hidden states $h_1^{lin}, \ldots, h_T^{lin} \in \mathbb{R}^{d_h}$ according to

$$h_t^{lin} = W h_{t-1}^{lin} + U x_t + b_i. \tag{7}$$

The output is the vector $V\sigma(h_T^{lin}) + b_o \in \mathbb{R}^{d_{out}}$.

- We call *ReLU-ORNNs*, those computing a sequence of hidden states $h_1^{ReLU}, \ldots, h_T^{ReLU} \in \mathbb{R}^{d_h}$ according to

$$h_t^{ReLU} = \sigma(Wh_{t-1}^{ReLU} + Ux_t + b_i).\qquad(8)$$

The output is the vector $Vh_T^{ReLU} + b_o \in \mathbb{R}^{d_{out}}$.

Notice that, for simplicity, we only consider many-to-one RNNs.

## G.1 Ablation study

We compare the performance of HadamRNN with linear recurrent units, as described in the article, (1) and (7), with those of a HadamRNN with ReLU recurrent units, in (8), which we refer to as HadamRNN-ReLU. The results are summarized in Table 8. It can be observed that the HadamRNN-ReLU fails to learn the copy task, and obtains poor results on the other benchmarks, as compared to the HadamRNN. This is because HadamRNN better memorize than HadamRNN-ReLU. This phenomenon is analyzed in Appendix G.2.

Table 8: Ablation study: performance comparison between HadamRNNs with linear recurrent units and with classical ReLU activation function recurrent unit. All results are presented with the size $d_h$ reported in Table 2 and 4-bit quantization for $U$ and $V$ matrices. BL (baseline) means that the model failed to learn.

| Model | Copy-task | pMNIST | sMNIST | IMDB |
|---|---|---|---|---|
| HadamRNN | 1.6e-7 | 94.88 | 96.63 | 87.43 |
| HadamRNN-ReLU | BL | 86.82 | 65.06 | 72.68 |

## G.2 Better memorization of linear recurrent units

### G.2.1 Theoretical arguments

In this section, we analyze first for linear-ORNNs, then for ReLU-ORNNs, the impact on the result of a modification of the input $x_t$, for $t \in [\![1, T]\!]$. To do so, we first compute $\frac{\partial h_T^{lin}}{\partial x_t}$ and then compute $\frac{\partial h_T^{ReLU}}{\partial x_t}$ and compare the behavior of the two quantities.

**The linear-ORNN case** We consider an input sequence $\mathbf{x} = (x_1, \ldots, x_T) \in \mathbb{R}^{d_{in} \times T}$, $t \in [\![1, T]\!]$ and $\mathbf{z} = (0, \ldots, 0, z_t, 0, \ldots, 0) \in \mathbb{R}^{d_{in} \times T}$. We denote $h_T^{lin}$ the last hidden state computed at $\mathbf{x}$ and $g_T^{lin}$ the one computed at $\mathbf{x} + \mathbf{z}$. We can show by induction that

$$g_T^{lin} = h_T^{lin} + W^{T-t}Uz_t.\qquad(9)$$

The impact of the perturbation $\mathbf{z}$ is independent of $\mathbf{x}$. The timestep $t$ influences $W^{T-t}$ but has no impact on the norm of the variation since, as $W$ is orthogonal,

$$\|W^{T-t}Uz_t\| = \|Uz_t\|.$$

In conclusion, a linear-ORNN retains information regardless of when the information occurs in the input sequence.

**The ReLU-ORNN case** Again, we consider an input sequence $\mathbf{x} = (x_1, \ldots, x_T) \in \mathbb{R}^{d_{in} \times T}$, $t \in [\![1, T]\!]$ and $\mathbf{z} = (0, \ldots, 0, z_t, 0, \ldots, 0) \in \mathbb{R}^{d_{in} \times T}$. We denote $h_T^{ReLU}$ the last hidden state computed at $\mathbf{x}$ and $g_T^{ReLU}$ the one computed at $\mathbf{x} + \mathbf{z}$. Due to the non-linear activation function, it is not possible to derive a straightforward formula analogous to (9). To analyze the behavior of $g_T^{ReLU}$, we use its first-order Taylor expansion and analyze its Jacobian.

For $t \in \{1, \dots, T\}$, the Jacobian[4] of $h_T^{ReLU}$ with respect to $x_t$ writes

$$\frac{\partial h_T^{ReLU}}{\partial x_t} = \frac{\partial h_T^{ReLU}}{\partial h_{T-1}^{ReLU}} \cdots \frac{\partial h_{t+2}^{ReLU}}{\partial h_{t+1}^{ReLU}} \frac{\partial h_{t+1}^{ReLU}}{\partial x_t}$$

$$= D_T W \cdots D_{t+1} W D_t U \in \mathbb{R}^{d_h \times d_{in}},$$

where, for all $s$, $D_s = \text{diag}\left(\sigma'\left(W h_{s-1} + U x_s + b_i\right)\right) \in \{0, 1\}^{d_h \times d_h}$ is diagonal.

The Euclidean norm of a vector is preserved when multiplied by $W$ but most often decreases when multiplied by a matrix $D_s$, since the latter often contains zeros on its diagonal. As a result, we expect the influence of variations of $x_t$, for $t$ small, to diminish or become negligible. Considering $x_t$ as a variation of 0, we see that the first inputs may have less influence on the result than the later ones.

### G.2.2 EXPERIMENTAL RESULTS

We conducted an experiment to empirically demonstrate that linear-ORNNs have better memory retention than ReLU-ORNNs.

**Setup** We construct a HadamRNN, a linear-ORNN, and consider the ReLU-ORNN obtained using the same weight matrices. They differ solely in the position of activation function (see (7) and (8)). For given inputs $x_1, \dots, x_T \in \mathbb{R}^{d_{in}}$, we denote the hidden state at time $t$ as $h_t^{lin}$ and $h_t^{ReLU}$, for $t = 1, \dots, T$.

Our aim is to empirically observe that, regardless of $t$, a perturbation in the $t$-th input has a consistent impact on the final hidden state $h_T^{lin}$. In contrast, the impact of the same perturbation on the final hidden state $h_T^{ReLU}$ of the ReLU-ORNN diminishes as $t$ decreases. As the time difference increases, the model gradually loses information .

To this end, we generate a 2-dimensional time series $(x_0, \dots, x_T) \in \mathbb{R}^{2 \times T}$, with $T = 200$ timesteps. We set $x_0 = (0, 0)$ and for $t = 1, \dots, T$, we sample $x_t$ according to the Gaussian distribution $\mathcal{N}\left(x_{t-1}, I_2\right)$.

We set $d_h = 128 = 2^7$, and we let the Hadamard recurrent weight matrix be $W = \frac{1}{\sqrt{d_h}} \mathbf{S}_{2^7}$, as in (3). The components of the input matrix $U \in \mathbb{R}^{128 \times 2}$ are independently sampled from the normal distribution $\mathcal{N}(0, 1)$. Since our primary focus is on how input variations affect the hidden state, we omit the output layer in our model.

**The experiment** To compute $\mathbf{e}^{lin}$ and $\mathbf{e}^{ReLU} \in \mathbb{R}^T$, we apply a perturbation to $x_t$ along the first axis, for all $t = 1, \dots, T$, and compute

$$\begin{cases} \mathbf{e}_t^{lin} = \|h_T^{lin}(x_1, \dots, x_t, \dots, x_T) - h_T^{lin}(x_1, \dots, x_t + (1,0), \dots, x_T)\|_2, \\ \mathbf{e}_t^{ReLU} = \|h_T^{ReLU}(x_1, \dots, x_t, \dots, x_T) - h_T^{ReLU}(x_1, \dots, x_t + (1,0), \dots, x_T)\|_2. \end{cases} \tag{10}$$

To compute $\mathbf{f}^{lin}$ and $\mathbf{f}^{ReLU} \in \mathbb{R}^T$, we apply the same perturbation along the second axis, and compute

$$\begin{cases} \mathbf{f}_t^{lin} = \|h_T^{lin}(x_1, \dots, x_t, \dots, x_T) - h_T^{lin}(x_1, \dots, x_t + (0,1), \dots, x_T)\|_2, \\ \mathbf{f}_t^{ReLU} = \|h_T^{ReLU}(x_1, \dots, x_t, \dots, x_T) - h_T^{ReLU}(x_1, \dots, x_t + (0,1), \dots, x_T)\|_2. \end{cases} \tag{11}$$

We plot the corresponding curves in Figure 3. As predicted by the analysis of Appendix G.2.1, for the linear-ORNN, the magnitude of the difference in hidden states remains constant, whether the perturbation is applied along the first or second axis. Conversely, in the ReLU-ORNN case, the norm increases, possibly approaching 0. This illustrates the fact that variations in $x_t$ have a diminishing impact on the hidden state $h_T$ as $T - t$ increases.

---

[4]For clarity, we use partial derivative notation for Jacobian matrices when applying the chain rule. It is important to note that while ReLU is not differentiable at 0 in the traditional sense, the calculations remain rigorous (Bolte and Pauwels, 2021).

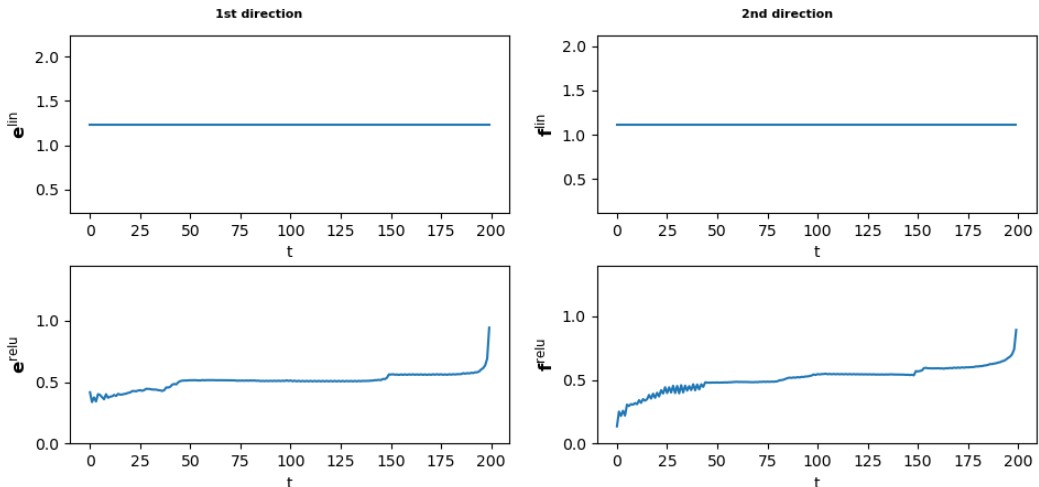

Figure 3: Top: $\mathbf{e}^{lin}$ and $\mathbf{f}^{lin}$, bottom: $\mathbf{e}^{ReLU}$ and $\mathbf{f}^{ReLU}$, see (10) and (11).

**Conclusion**   Linear-ORNNs have better memorization property than ReLU-ORNNs, and are able to retain information over arbitrarily long delays. This information may be lost after a certain delay with a ReLU-ORNN.

# H   SWITCHING COLUMNS SIGNS IS NOT USEFUL

## H.1   SWITCHING COLUMNS SIGNS DOES NOT PROVIDE MORE EXPRESSIVENESS

In this section, we first provide a first result stating that modifying the signs of the lines of a Hadamard matrix (as is done by the multiplication by $\mathrm{diag}(u)$, in Proposition 3.3 and the definition of the $W(u)$s in (3) and (4)) defines the same hidden states as switching the signs of the columns. We then state that changing the signs of both rows and columns still defines the same hidden states and is therefore unnecessary.

To do so, we define, for $n$ and $d_h \in \mathbb{N}^*$ and a given matrix $H \in \mathbb{R}^{d_h \times n}$, the operator $\phi_H$ by:

$$\phi_H : \{-1, 1\}^{d_h} \longrightarrow \mathbb{R}^{d_h \times n}$$
$$u \longmapsto \mathrm{diag}(u)H,$$

and, for $H \in \mathbb{R}^{n \times d_h}$, the operator $\Gamma_H$ by

$$\Gamma_H : \{-1, 1\}^{d_h} \longrightarrow \mathbb{R}^{n \times d_h}$$
$$u \longmapsto H \, \mathrm{diag}(u).$$

**Proposition H.1.** *Let $H \in \mathbb{R}^{d_h \times d_h}$, $U \in \mathbb{R}^{d_h \times d_{in}}$, $V \in \mathbb{R}^{d_{out} \times d_h}$ and $b_i \in \mathbb{R}^{d_h}$.*

*For all $u \in \{-1, 1\}^{d_h}$, the RNNs states $h_t$ and $h_t'$ defined for all $x_1, \ldots, x_T \in \mathbb{R}^{d_{in}}$ by*

$$\begin{cases} h_t = \phi_H(u)h_{t-1} + Ux_t + b_i & \text{for all } t \in [\![1, T]\!] \\ h_t' = \Gamma_H(u)h_{t-1}' + \phi_U(u)x_t + \phi_{b_i}(u) & \text{for all } t \in [\![1, T]\!] \end{cases}$$

*with $h_0 = h_0' = 0$, differ only by their sign:*

$$h_t' = \mathrm{diag}(u)h_t, \text{ for all } t \in [\![1, T]\!].$$

*Proof.* Consider $H \in \mathbb{R}^{d_h \times d_h}$, $U \in \mathbb{R}^{d_h \times d_{in}}$, $V \in \mathbb{R}^{d_{out} \times d_h}$ and $b_i \in \mathbb{R}^{d_h}$. Let $x_1, \ldots, x_T \in \mathbb{R}^{d_{in}}$.

Let us first prove that for all $u \in \{-1, 1\}^{d_h}$ and all $t \in [\![1, T]\!]$, $h_t' = \mathrm{diag}(u)h_t$. Let us consider $u$ and prove that the result holds by induction.

Table 9: Comparison of the performances of the linear HadamRNN in which only the rows switches are learned, and the linear HadamRNN in which both rows and columns switches are learned. $W$ is binary, and $U$ and $V$ are quantized over 4 bits.

| Model | row switches | row and column switches |
|---|---|---|
| copy task | 1.6e-7 | 1.3e-8 |
| Permuted MNIST | 94.88 | 28.99 |
| Sequential MNIST | 96.63 | 96.31 |
| IMDB | 87.43 | 85.24 |

- **Initialization:** We have $h'_0 = 0 = \text{diag}(u)0 = \text{diag}(u)h_0$.

- **Heredity:** Assume $t \geq 1$ is such that $h'_{t-1} = \text{diag}(u)h_{t-1}$.

  Using the definition of $h'_t$, the fact that $\text{diag}(u)\text{diag}(u) = I_{d_h}$ and the definition of $h_t$, we have

$$
\begin{aligned}
h'_t &= \Gamma_H(u)h'_{t-1} + \phi_U(u)x_t + \Phi_{b_i}(u) \\
&= H\,\text{diag}(u)\,\text{diag}(u)h_{t-1} + \text{diag}(u)Ux_t + \text{diag}(u)b_i \\
&= \text{diag}(u)\left(\text{diag}(u)Hh_{t-1} + Ux_t + b_i\right) \\
&= \text{diag}(u)h_t
\end{aligned}
$$

This concludes the proof of the proposition. □

This proposition ensures that, when optimizing $U$ and $b_i$ over a set that is invariant under multiplication by $\text{diag}(u)$ (whether on the left or right), choosing left or right yields equivalent hidden states.

Applying Proposition H.1 to $H' = \text{diag}(u')H$, for a fixed $u' \in \{-1, 1\}^{d_h}$, shows that changing the signs of both rows and columns results in the same hidden state as changing the signs of the rows only.

### H.2 SWITCHING COLUMNS SIGNS ARMS RESULTS

We compared a version of our linear HadamRNN where both row and column switches are learned during training, to the standard linear HadamRNN in which only row switches are learned, as outlined in this paper.

The two versions of the models and the datasets use the same configurations as detailed in Section 4 and Appendix D. We used a binary orthogonal recurrent weight matrix for $W$ (see Section 3.3) and quantized $U$ and $V$ over 4 bits. Activations were not quantized.

Results are reported in Table 9. The standard version consistently outperforms the variant, except in the copy task. This finding, along with Appendix H.1, supports our choice to learn row switches only.

## I POST TRAINING QUANTIZATION OF ACTIVATIONS

### I.1 REMINDERS ON FIXED-POINT ARITHMETIC

We follow conventional notations for fixed-point arithmetic: For integers $q \geq 0$ and $p \geq 1$, the set of $p$-bit fixed-point numbers with $q$ bits allocated for the fractional part is represented as:

$$
\mathcal{Q}_{p,q} = \frac{1}{2^q}[\![-2^{p-1}, 2^{p-1} - 1]\!] \subset \left[-2^{p-q-1}, 2^{p-q-1}\right) \subset \mathbb{R}.
$$

In the specific case where $p = q + 1$, we use the simplified notation:

$$
\mathcal{Q}_q = \mathcal{Q}_{q+1,q} = \frac{1}{2^q}[\![-2^q, 2^q - 1]\!] \subset [-1, 1).
$$

The addition of two fixed-point numbers is only possible when they have identical fractional sizes. For example, if $x$ and $x' \in \mathcal{Q}_q$, then $x + x' \in \mathcal{Q}_{q+2,q}$.

Multiplying two fixed-point numbers $x \in \mathcal{Q}_{p,q}$ and $x' \in \mathcal{Q}_{p',q'}$ results in a value within $x \cdot x' \in \mathcal{Q}_{p+p'-1,q+q'}$. Thus, the product of $x \in \mathcal{Q}_q$ and $x' \in \mathcal{Q}_{q'}$ satisfies $x \cdot x' \in \mathcal{Q}_{q+q'+1,q+q'} = \mathcal{Q}_{q+q'}$.

## I.2  Weight quantization:

In the description of HadamRNN and Block-HadamRNN, matrices $U \in \mathbb{R}^{d_h \times d_{in}}$ and $V \in \mathbb{R}^{d_{out} \times d_h}$ are quantized on $p$ bits as described in Section 3.5. We will note $q_p(U) = \alpha_U \widetilde{U}$ (resp. $q_p(V) = \alpha_V \widetilde{V}$) where the quantized matrix $\widetilde{U} \in \mathcal{Q}_{p-1}^{d_h \times d_{in}} = \mathcal{Q}_{p,p-1}^{d_h \times d_{in}}$ (resp. $\widetilde{V} \in \mathcal{Q}_{p-1}^{d_{out} \times d_h}$), and $\alpha_U = \|U\|_{\max} > 0$ (resp. $\alpha_V = \|V\|_{\max} > 0$), where $\|U\|_{\max} = \sup_{i,j} |U_{i,j}|$.

Given the definition of $W$ in (3) and (4), we note the recurrent matrix $W = \alpha_W \widetilde{W}$, with $\widetilde{W} \in \mathcal{Q}_1^{d_h \times d_h}$ and $\alpha_W = \frac{2}{\sqrt{d_h}}$ for HadamRNN (or $\alpha_W = \frac{2}{\sqrt{2^k}}$ for Block-HadamRNN). Notice indeed that, since $\mathcal{Q}_1 = \{-1, \frac{1}{2}, 0, \frac{1}{2}\}$, all the matrices $W$ in $\frac{1}{\sqrt{d_h}}\{-1, +1\}^{d_h \times d_h}$ (resp. $\frac{1}{\sqrt{2^k}}\{-1, 0, +1\}^{d_h \times d_h}$) are also in $\frac{2}{\sqrt{d_h}}\mathcal{Q}_1^{d_h \times d_h}$ (resp. $\frac{2}{\sqrt{2^k}}\mathcal{Q}_1^{d_h \times d_h}$). Taking $\mathcal{Q}_0 = \{-1, 0\}$ instead of $\mathcal{Q}_1$ does not permit the components of $\frac{2}{\sqrt{d_h}}\widetilde{W}$ (resp $\frac{2}{\sqrt{2^k}}\widetilde{W}$) to reach $+\frac{1}{\sqrt{d_h}}$ (resp $+\frac{1}{\sqrt{2^k}}$).

## I.3  Input and Hidden State encoding:

For each time step $t \in [\![1, T]\!]$, the quantized hidden state $h_t$ is encoded using $p_a$ bits, where $p_a \geq 1$. A fixed scaling factor $\alpha_h > 0$ is applied such that $h_t = \alpha_h \tilde{h}_t$, with $\tilde{h}_t \in \mathcal{Q}_{p_a-1}^{d_h}$. We use the notation $q_{p_a}^{\alpha_h}(x)$ to denote the closest element of $x \in \mathbb{R}$ in $\alpha_h \mathcal{Q}_{p_a-1}$. This notation extends to vectors.

In practice, $\alpha_h$ must be sufficiently large to cover the range of values for the full-precision hidden states. However, increasing $\alpha_h$ slightly may not significantly impact performance, allowing some flexibility in its selection.

Similarly, each input $x_t$, for $t \in [\![1, T]\!]$, is quantized using $p_i$ bits, with $p_i \geq 1$. For simplicity, we continue to denote the quantized inputs as $x_t$. Using a fixed scaling factor $\alpha_i > 0$, we represent $x_t = \alpha_i \tilde{x}_t$, where $\tilde{x}_t \in \mathcal{Q}_{p_i-1}^{d_{in}}$.

**Input Quantization Examples:**

- For the copy-task where input entries are either 0 or 1, we use $\alpha_i = 2$. As long as $p_i \geq 2$, quantization does not alter the inputs.

- For the pixel-by-pixel MNIST tasks, with normalized 8-bit unsigned integer values in $[0, 1]$, we set $\alpha_i = 1$. Quantization has no effect as long as $p_i \geq 9$.

- For the IMDB dataset, the 512 inputs are given by a floating point word embedding preprocessing. We quantize these preprocessing outputs using $p_i = p_a$ bits (i.e. within $\mathcal{Q}_{p_a-1}$), and set $\alpha_i$ to the maximum value of the embeddings.

## I.4  Independence to scaling factors

It can be shown by induction that a vanilla (Linear or ReLU) RNN with parameters $(W, b_i, U, V, b_o)$ produces the same outputs as an RNN with parameters $(W, \lambda b_i, \lambda U, \frac{1}{\lambda}V, b_o)$ for all $\lambda > 0$.

In the following sections, we use this idea and, instead of applying the network with quantized weights $(q_1(W), b_i, q_p(U), q_p(V), b_o) = (\alpha_W \widetilde{W}, b_i, \alpha_U \widetilde{U}, \alpha_V \widetilde{V}, b_o)$, we rescale with $\lambda = \frac{1}{\alpha_i \alpha_U}$ and apply the network with the parameters $(\alpha_W \widetilde{W}, \frac{b_i}{\alpha_i \alpha_U}, \frac{1}{\alpha_i} \widetilde{U}, \alpha_i \alpha_U \alpha_V \widetilde{V}, b_o)$.

Table 10: Value of $\alpha_W$ and $\alpha_h$ for activation quantification across the datasets and bitwidth.

| Model | U,V bitwidth | Copy-task | | sMNIST | | pMNIST | | IMDB | |
|---|---|---|---|---|---|---|---|---|---|
| | | $\alpha_W$ | $\alpha_W \alpha_h$ | $\alpha_W$ | $\alpha_W \alpha_h$ | $\alpha_W$ | $\alpha_W \alpha_h$ | $\alpha_W$ | $\alpha_W \alpha_h$ |
| HadamRNN | 4 | 0.088 | 2.0 | 0.044 | 4.0 | 0.044 | 1.0 | 0.044 | 8.0 |

## I.5 RECURRENCE WITH FIXED-POINT ARITHMETIC :

Given, the quantized hidden state $h_t = \alpha_h \tilde{h}_t \in \alpha_h Q_{p_a-1}^{d_h}$, with a fixed value of $\alpha_h$ chosen later, and the quantized input $x_t = \alpha_i \tilde{x}_t \in \alpha_i Q_{p_i-1}^{d_{in}}$.

The recurrence relation defined in (1) for the parameters $(\alpha_W \widetilde{W}, \frac{b_i}{\alpha_i \alpha_U}, \frac{1}{\alpha_i}\widetilde{U}, \alpha_i \alpha_U \alpha_V \widetilde{V}, b_o)$ becomes:

$$\alpha_h \tilde{h}_t = h_t = q_{p_a}^{\alpha_h} \left( \alpha_W \widetilde{W} h_{t-1} + \frac{1}{\alpha_i}\widetilde{U} x_t + \frac{b_i}{\alpha_i \alpha_U} \right)$$

$$= q_{p_a}^{\alpha_h} \left( \alpha_W \alpha_h \widetilde{W} \tilde{h}_{t-1} + \widetilde{U} \tilde{x}_t + \frac{b_i}{\alpha_i \alpha_U} \right).$$

In this equation, the matrix-vector multiplications $\widetilde{W}\tilde{h}_{t-1}$ and $\widetilde{U}\tilde{x}_t$ are computed using fixed-point arithmetic.

We take advantage of the flexibility in choosing $\alpha_h$ to ensure that multiplying by $\alpha_W \alpha_h$ can be performed with a simple bit shift. We first compute $\max_h = \max_{t \in [\![1,T]\!]} \|h_t\|_\infty$, based on the full-precision hidden states from the training and validation datasets. We choose $n_h = argmin_{n \in \mathbb{N}}\{2^n \geq \max_h \alpha_W\}$, and set $\alpha_h = 2^{n_h}/\alpha_W$.

The values used in the experiments are presented in Table 10.

We also quantize the scaled bias using $p_a$ bits.

Finally, the entry of the quantization $q_{p_a}^{\alpha_h}$ is a fixed-point value and thus belong to a finite set, the size of which depends on $\alpha_W \alpha_h$ and $(p, p_a, p_i)$. Thus, $\tilde{h}_t$ can be computed using a simple look-up table, eliminating the need for floating-point operations (division by $\alpha_h$ in $q_{p_a}^{\alpha_h}$).

The quantization of the output layer follows the classical PTQ methodology, using a quantized value of the bias $b_o$ and the multiplicative factor $\alpha_i \alpha_U \alpha_V$.

This leads to a fully quantized RNN. The Table 2 shows that such fully quantized RNNs with $p_a = 12$ bits achieve equivalent results than the RNNs with quantized weights only.

