# OpenReview forum: "HADAMRNN: BINARY AND SPARSE TERNARY ORTHOGONAL RNNS"
_ICLR.cc/2025/Conference — ICLR 2025 Poster_

### Official Review · Reviewer_wQjw · 2024-10-18

**Soundness:** 3
**Presentation:** 3
**Contribution:** 3
**Rating:** 6
**Confidence:** 4

**Summary:**

This paper tackles the challenge of binarizing orthogonal recurrent neural networks (oRNNs), a problem that has been hindered by instability and inconsistency in previous formulations. The authors propose HadamRNN, a novel approach that utilizes Hadamard matrices to parameterize oRNNs with binary and sparse ternary weights. This method cleverly addresses the instability issues and offers a consistent framework for binarizing oRNNs. Furthermore, the paper introduces an optimized block version of HadamRNN, which brings about computational efficiency.

**Strengths:**

1) Novel Approach: The introduction of Hadamard matrices for parameterization is a novel and effective way to address the instability issues associated with binarizing oRNNs.

2) Theoretical Soundness: The paper provides a solid theoretical foundation for the proposed method and demonstrates its consistency.

3) Performance: The empirical results on benchmark datasets (copy task, MNIST, IMDB) are promising, showing comparable performance to full-precision models. Notably, HadamRNN is the first binary recurrent weight model to handle the copy task over 1000 timesteps.

4) Optimization: The optimized block version of HadamRNN offers improved computational efficiency.


--------------------------------------
After rebuttal:

The authors have addressed my concerns at length and added addiitonal experiments supporting the claims. I am in support of accepting the paper but might not be able to champion it.

**Weaknesses:**

At a high-level, while I appreciate the work towards making oRNNs work better and efficient, they themselves have quite a few issues including expressivity.

1) Limited Real-World Applicability: While the benchmarks used are common in the literature, they often do not translate well to real-world scenarios. oRNNs, in general, have not seen widespread adoption in practical applications.

2) Training Time: The authors mention FastGRNN (Kusupati et al., 2018), which suggests that training oRNN models can be significantly slower (an order of magnitude or more) compared to other methods. It would be beneficial to see a comparison of training times for the proposed HadamRNN models against relevant baselines. Additionally, a discussion on the stability of training these models would be valuable.

3) Lack of Diverse Datasets: The evaluation primarily relies on toy datasets. To strengthen the claims and demonstrate broader applicability, it would be crucial to include more real-world datasets, even if smaller scale, such as those from the GLUE benchmark (e.g., SST-2, QQP) or SQUAD for question answering.

**Questions:**

At this point I am on the border with a leaning towards rejection but the following things might help along with other reviewers comments.

The paper presents a valuable contribution to the field of oRNNs by introducing a stable and efficient binarization method. However, the lack of real-world evaluation and potential concerns regarding training time and stability hinder its immediate acceptance. I encourage the authors to address these concerns by:

1) Expanding the evaluation to include more diverse and realistic datasets. This would demonstrate the practical value of HadamRNN and provide a more comprehensive assessment of its capabilities.

2) Providing a detailed analysis of training time and stability. This would offer valuable insights into the practical considerations of deploying HadamRNN in real-world applications.

By addressing these points, the authors can significantly strengthen the paper and increase its impact.

---

> ### Author Response · Authors · 2024-11-24
> **Addressing weaknesses and questions raised by reviewer wQjw**
>
> **Weaknesses:**
>
> **At a high-level, while I appreciate the work towards making oRNNs work better and efficient, they themselves have quite a few issues including expressivity.**
>
> **Limited Real-World Applicability: While the benchmarks used are common in the literature, they often do not translate well to real-world scenarios. oRNNs, in general, have not seen widespread adoption in practical applications.**
>
> We provide an answer to this issue below, where we describe the new benchmarks that we consider.
>
> **Training Time: The authors mention FastGRNN (Kusupati et al., 2018), which suggests that training oRNN models can be significantly slower (an order of magnitude or more) compared to other methods. It would be beneficial to see a comparison of training times for the proposed HadamRNN models against relevant baselines. Additionally, a discussion on the stability of training these models would be valuable**
>
>  We provide an answer to this issue below.
>
>
> **Lack of Diverse Datasets: The evaluation primarily relies on toy datasets. To strengthen the claims and demonstrate broader applicability, it would be crucial to include more real-world datasets, even if smaller scale, such as those from the GLUE benchmark (e.g., SST-2, QQP) or SQUAD for question answering.**
>
> We provide an answer to this issue below, where we describe the new benchmarks that we consider.
>
>
>
>
> **Questions:**
>
> **At this point I am on the border with a leaning towards rejection but the following things might help along with other reviewers comments.**
>
> **The paper presents a valuable contribution to the field of oRNNs by introducing a stable and efficient binarization method. However, the lack of real-world evaluation and potential concerns regarding training time and stability hinder its immediate acceptance. I encourage the authors to address these concerns by:**
>
> **Expanding the evaluation to include more diverse and realistic datasets. This would demonstrate the practical value of HadamRNN and provide a more comprehensive assessment of its capabilities.**
>
> Our primary focus was on benchmarks widely used within the RNN, ORNN, and LSTM research communities. In addition, we included the IMDB dataset, a realistic sentiment analysis task in natural language processing (NLP). To further showcase the versatility and strong performance of HadamRNN and Block-HadamRNN across diverse scenarios, we introduced several additional benchmarks, described in detail below.
>
> From GLUE, as requested, we include at least the SST-2 benchmark. This provides also the opportunity to compare HadamRNN to binary and ternary transformers, as asked by reviewer gcYV. On SST-2, we observe that HadamRNN models with sizes of just a few kilobytes achieve an accuracy exceeding 80\% (we expect to reach 82–83\%). While this is lower than the performance of transformers, which achieve higher accuracy using pre-trained weights and have sizes in the tens of megabytes, it highlights the efficiency of HadamRNN in terms of model size.
>
> Still from the GLUE benchmark, we are working on obtaining experimental results for the CoLA benchmark. However, these results will only be included in the article if they are ready for publication.
>
> To demonstrate the applicability of HadamRNN and Block-HadamRNN, we also include comparisons on the HAR-2, DSA-19, and (hopefully) Yelp-5 benchmarks, as presented in the FastGRNN article (1). The HAR-2 and DSA-19 datasets are designed for recognizing human activity from sensor measurements, while Yelp-5 is a sentiment analysis dataset based on text.
>
> Because of space limitation, all the new benchmarks are in Appendix.
>
> 1. Kusupati, Aditya, et al. Fastgrnn: A fast, accurate, stable and tiny kilobyte-sized gated recurrent neural network. Advances in neural information processing systems 31 (2018). [https://arxiv.org/pdf/1901.02358](https://arxiv.org/pdf/1901.02358)
>
>
>
> **Providing a detailed analysis of training time and stability. This would offer valuable insights into the practical considerations of deploying HadamRNN in real-world applications.**
>
> In the Appendix, we provide a table that contains the training time of the models for each dataset.
>
> We also provide a table for the IMDB dataset that shows, for different values of $p$ (the bit-width of $U$ and $V$) and various initial learning rates, the average performance and standard deviation across different random seeds, where the seed influences both the initial weights and the stochastic optimizer.
>
> **By addressing these points, the authors can significantly strengthen the paper and increase its impact.**
>
> We hope that these additional results will demonstrate the potential of this paper.

---

> > ### Comment · Reviewer_wQjw · 2024-11-25
> >
> > Thanks for the rebuttal. I can't seem to find the updated version of the paper to check the appendix. Let me know if I am missing something.

---

> > > ### Author Response · Authors · 2024-11-25
> > > **Official comment by authors**
> > >
> > > You are correct, we did not upload the updated version yet. We will upload it on november 26.

---

### Official Review · Reviewer_gcYV · 2024-11-02

**Soundness:** 2
**Presentation:** 3
**Contribution:** 3
**Rating:** 6
**Confidence:** 4

**Summary:**

Binary and sparse ternary weights in neural networks enable faster computations and lighter representations, facilitating their use on edge devices with limited computational power. However, to date, no method has successfully achieved binarization or ternarization of vanilla Recurrent Neural Network (RNN) weights. As a result, in this paper, the authors present a new approach leveraging the properties of Hadamard matrices to parameterize a subset of binary and sparse ternary orthogonal matrices. The method enables the training of Orthogonal RNNs (ORNNs) with binary and sparse ternary recurrent weights, effectively creating a specific class of binary and sparse ternary vanilla RNNs. Despite binarization or sparse ternarization, the proposed RNNs maintain performance levels comparable to state-of-the-art full-precision models, highlighting the approach's effectiveness.

**Strengths:**

The paper is well-organized and has a high level of clarity in its writing and the presentation of its proposal.

Each section flows logically, enhancing the reader's comprehension of the paper's purpose and methodology. Notably, sections 3 (+ the Appendix) stand out for its depth and precision, guiding the reader through the formality of the proposed approach and leaving little room for ambiguity regarding the study's methods and objectives.

The proposal is interesting. Although it is presented explicitly for RNN, the underlying principles and approaches have strong adaptability potential. The proposed sparse ternary orthogonal recurrent weights could be effectively applied across various domains (with necessary modifications for the new application domain).

**Weaknesses:**

My major concern with this paper is regarding the experimental section. Specifically, in lines 123 - 125, the authors claim: *"Even with binarization, transformer models [...], making them unsuitable for tasks involving long-term dependencies on edge devices."* I find this assertion to be overly restrictive, as recent literature provides a substantial body of work on optimizing Transformers for edge devices and handling long-term dependencies, which is not discussed here. For example, numerous studies have been conducted on lightweight binary Transformers [1, 2, 3, 4] and approaches specifically designed to handle long-term dependencies [5, 6, 7].
Consequently, the state-of-the-art section also appears narrow, as it overlooks several relevant studies and advancements in Transformer models that are critical to this discussion.

[1] Liu, Z., Oguz, B., Pappu, A., Xiao, L., Yih, S., Li, M., ... & Mehdad, Y. (2022). BiT: Robustly Binarized Multi-distilled Transformer. Advances in Neural Information Processing Systems (NeurIPS), 35, 14303-14316.

[2] He, Y., Lou, Z., Zhang, L., Liu, J., Wu, W., Zhou, H., & Zhuang, B. (2023). BiViT: Extremely Compressed Binary Vision Transformers. In Proceedings of the IEEE/CVF International Conference on Computer Vision (pp. 5651-5663).

[3] Wang, Z., Luo, H., Xie, X., Wang, F., & Shi, G. (2024, March). BVT-IMA: Binary Vision Transformer with Information-Modified Attention. In Proceedings of the AAAI Conference on Artificial Intelligence (Vol. 38, No. 14, pp. 15761-15769).

[4] Gao, T., Xu, C. Z., Zhang, L., & Kong, H. (2024). GSB: Group superposition binarization for vision transformer with limited training samples. Neural Networks, 172, 106133.

[5] Wu, H., Xu, J., Wang, J., & Long, M. (2021). Autoformer: Decomposition Transformers with Auto-Correlation for Long-Term Series Forecasting. Advances in Neural Information Processing Systems (NeurIPS), 34, 22419-22430.

[6] Zhou, H., Zhang, S., Peng, J., Zhang, S., Li, J., Xiong, H., & Zhang, W. (2021, May). Informer: Beyond Efficient Transformer for Long Sequence Time-Series Forecasting. In Proceedings of the AAAI conference on artificial intelligence (Vol. 35, No. 12, pp. 11106-11115).

[7] Zhou, T., Ma, Z., Wen, Q., Wang, X., Sun, L., & Jin, R. (2022, June). FEDformer: Frequency Enhanced Decomposed Transformer for Long-term Series Forecasting. In International conference on machine learning (pp. 27268-27286). PMLR.

**Questions:**

Summarizing: The proposal is interesting. However, the weaknesses presented in terms of state-of-the-art and comparison are not so minor. Therefore, this work needs more effort to be improved, and it is not ready for publication in this state.

Based on the above, expanding the experiments and the related work to include additional competitors from Transformer-based models would significantly strengthen the paper's relevance and situate it better within the current research landscape.

Other minor considerations: I recommend that the authors carefully re-read the work; there are several typos and English errors. Furthermore, the plots in the Appendix do not specify what the X-axis and Y-axis represent.

---

> ### Author Response · Authors · 2024-11-24
> **Strengths - on the versatility of our method**
>
> **The paper is well-organized and has a high level of clarity in its writing and the presentation of its proposal.**
>
> **Each section flows logically, enhancing the reader's comprehension of the paper's purpose and methodology. Notably, sections 3 (+ the Appendix) stand out for its depth and precision, guiding the reader through the formality of the proposed approach and leaving little room for ambiguity regarding the study's methods and objectives.**
>
> **The proposal is interesting. Although it is presented explicitly for RNN, the underlying principles and approaches have strong adaptability potential. The proposed sparse ternary orthogonal recurrent weights could be effectively applied across various domains (with necessary modifications for the new application domain).**
>
> Indeed, the article focuses on the use of Hadamard matrices for RNNs, but their potential applications extend beyond this scope.
>
> As a perspective, we outline a list of potential applications that could also benefit from similar Hadamard matrices, accompanied by bibliographical references for each. These applications of orthogonal layers include long-term time series forcasting (the three references provided by the reviewer), provable robustness of neural networks[1,2], Normalizing Flows[3], Wassertein distance estimationfor instance for WGANs [4]. We add a sentence in the conclusion.
>
> 1. Cisse, Moustapha and Bojanowski, Piotr and Grave, Edouard and Dauphin, Yann and Usunier, Nicolas. Parseval networks: Improving robustness to adversarial examples. ICML'17.
> 2. Anil, Cem and Lucas, James and Grosse, Roger. Sorting out Lipschitz function approximation.ICML'19.
> 3. Kingma, Durk P and Dhariwal, Prafulla. Glow: Generative flow with invertible 1x1 convolutions.NeurIPS'18.
> 4. Brock, Andrew and Donahue, Jeff and Simonyan, Karen. Large Scale GAN Training for High Fidelity Natural Image Synthesis.ICLR'18.

---

> ### Author Response · Authors · 2024-11-24
> **Weaknesses**
>
> **My major concern with this paper is regarding the experimental section. Specifically, in lines 123 - 125, the authors claim: "Even with binarization, transformer models [...], making them unsuitable for tasks involving long-term dependencies on edge devices." I find this assertion to be overly restrictive, as recent literature provides a substantial body of work on optimizing Transformers for edge devices and handling long-term dependencies, which is not discussed here. For example, numerous studies have been conducted on lightweight binary Transformers [1, 2, 3, 4] and approaches specifically designed to handle long-term dependencies [5, 6, 7]. Consequently, the state-of-the-art section also appears narrow, as it overlooks several relevant studies and advancements in Transformer models that are critical to this discussion.**
>
>
> We acknowledge that lines 123–125 could be misinterpreted. We agree that these lines do
>  not accurately reflect the current state of the art. We will include an expanded list of references to articles discussing binarized, ternarized, and quantized transformers for time series and Natural Language Processing (NLP). Additionally, we will provide a comprehensive list of references to works focused on improving transformers' ability to handle long-term dependencies.
>
> Thank you very much for all the references.

---

> ### Author Response · Authors · 2024-11-24
> **Questions**
>
> **Summarizing: The proposal is interesting. However, the weaknesses presented in terms of state-of-the-art and comparison are not so minor. Therefore, this work needs more effort to be improved, and it is not ready for publication in this state.**
>
> **Based on the above, expanding the experiments and the related work to include additional competitors from Transformer-based models would significantly strengthen the paper's relevance and situate it better within the current research landscape.**
>
> As mentioned earlier, we will expand the bibliography related to transformers.
>
> In terms of experiments, we will include at least one additional benchmark where quantized transformers have been applied and present it in a new appendix.
>
> On the SST-2 benchmark from GLUE, as suggested by reviewer wQJw, HadamRNN models with sizes of just a few kilobytes achieve an accuracy exceeding 80\% (with a target of 82–83\%). While this is lower than the performance of transformers, which achieve higher accuracy using pre-trained weights and have sizes in the tens of megabytes, it highlights the efficiency of HadamRNN in terms of model size.
>
> We are also working on obtaining experimental results for the CoLA benchmark from GLUE. However, these results will only be included in the article if they are ready for publication.
>
>
> **Other minor considerations: I recommend that the authors carefully re-read the work; there are several typos and English errors.**
>
> We have carefully proofread the article, but if you have identified any specific typos or errors, we would greatly appreciate it if you could share them with us. Your feedback would be invaluable.
>
> **Furthermore, the plots in the Appendix do not specify what the X-axis and Y-axis represent.**
>
> We have labeled the axes of the plot.

---

> > ### Comment · Reviewer_gcYV · 2024-11-26
> >
> > Dear Authors, thank you for your responses and the effort you have put into addressing my feedback. I am inclined to raise my score based on your work thus far. However, I will take the final decision only after reviewing the updated version of the article with the new results and updates you reported during this discussion stage.

---

### Official Review · Reviewer_fwft · 2024-11-02

**Soundness:** 3
**Presentation:** 2
**Contribution:** 3
**Rating:** 6
**Confidence:** 2

**Summary:**

This paper proposes a method to parameterize the recurrent weights of Orthogonal Recurrent Neural Networks (ORNN), a special case of Vanilla RNN to binary or sparse ternary matrices. The binarization and sparse ternarization algorithms that the paper proposed are based on Hadamard Matrix Theory. The paper tests the algorithms on two types of binarized and sparse ternarized Orthogonal Vanilla Recurrent Networks and compares their performance to LSTMs and GRUs recurrent models for four standard benchmark datasets.

**Strengths:**

The paper proposes a novel approach to parameterize weights of Orthogonal RNN models using Hadamard Matrix Theory. As mentioned in the paper the binary and sparse ternary parameterization are investigated and analyzed on lightweight neural networks for time series, the results reported in Table 2 highlight the potential of the proposed approach by comparing to ORNN, LSTM and FastGRNN recurrent models.

**Weaknesses:**

The binarization and ternarization algorithms of Orthogonal Vanilla RNN are explained in sections 3.3 and 3.4 in a high-level way. Maybe providing a base example by giving the dimensions of the matrices to be binarized or ternarized would make the algorithms more transparent and easy to understand.

The results reported in Table 2 and Table 3 are tough to read. Maybe converting some of the results into plots would make it more straightforward for the reader to assess the paper's contributions. It is unclear where the advantage of using Hadamard Matrix Theory to parameterize the binary orthogonal recurrent weights lies .
The paper does not analyze the method proposed on any edge device. Reporting the overhead and the speedup obtained on HadamardRNN and Block-HadamardRNN would help assess the paper's contribution.

**Questions:**

- Could you evaluate the performance of Block-HadamRNN and HadamRNN on edge devices? In particular, would you be able to provide the speedup and computational overhead that while using your techniques on these devices?

- Could you maybe think about making Tables 2 and 3 simpler to improve the readability of your results? Better visual perception may also result from the inclusion of a plot that contrasts the accuracy of all the suggested and benchmark approaches.

- Could you maybe include a thorough example in the text that shows how a trained ORNN is subjected to the binarization or sparse ternarization algorithms? Could you also explain the Straight-Through Estimator's (STE) function in the parameterization process?

---

> ### Author Response · Authors · 2024-11-24
> **Weaknesses**
>
> **The binarization and ternarization algorithms of Orthogonal Vanilla RNN are explained in sections 3.3 and 3.4 in a high-level way. Maybe providing a base example by giving the dimensions of the matrices to be binarized or ternarized would make the algorithms more transparent and easy to understand.**
>
> We appreciate the reviewer’s suggestion to provide more clarity on the parameterization using Hadamard matrices. To address this, in the revised version of the manuscript, in Appendix C.2, we explicitly present the recurrent weight matrix for HadamRNN when $d_h = 4$.
>
> **The results reported in Table 2 and Table 3 are tough to read. Maybe converting some of the results into plots would make it more straightforward for the reader to assess the paper's contributions. It is unclear where the advantage of using Hadamard Matrix Theory to parameterize the binary orthogonal recurrent weights lies.**
>
> It is common practice to present results in tables, as demonstrated by the articles cited in the bibliography below. This approach offers the advantage of providing precise numerical values, which are often difficult to extract from plots, and detailed information about the experimental setup.
>
> We have therefore decided to retain the tables while providing additional plots in a new appendix, Appendix E.3. These plots illustrate how model size is reduced without compromising performance, with each model represented as a point on the plane. The coordinates of each point correspond to the model's size and its performance as reported in Table 2.
>
> We also illustrate, in the same appendix, the complexity reduction achieved by Block-HadamRNN compared to HadamRNN using similar plots, where the x-axis represents complexity and the y-axis continues to represent performance.
>
> **The paper does not analyze the method proposed on any edge device. Reporting the overhead and the speedup obtained on HadamardRNN and Block-HadamardRNN would help assess the paper's contribution.**
>
> Implementing HadamRNNs and Block-HadamRNNs on edge devices would indeed be valuable. However, this requires substantial effort and different skills. In general, in the literature, the binarization procedure and the implementation are presented in separate papers, particularly for binary neural networks that require specific hardware (for instance FPGA). Many research articles focus on methods for building quantized or binary neural networks and do not implement them on edge devices [1, 2, 3, 4, 5]. The article  implementing a quantized neural network on an edge device often are quite different in nature [6, 7].
>
> As demonstrated in the Appendix, inference can be efficiently carried out using only integer operations through fixed-point arithmetic, marking a significant advancement toward practical implementation. Furthermore, Tables 2 and 3 detail the sizes of the neural networks and their computational complexities (measured in terms of additions and multiplications), providing a reliable estimate of their performance on edge devices.
>
> We consider the implementation of HadamRNNs and Block-HadamRNNs on edge devices as a future direction of research and mention it as a perspective in Section 5.
>
>
>
> 1. Joachim Ott, Zhouhan Lin, Ying Zhang, Shih-Chii Liu, and Yoshua Bengio. Recurrent neural networks with limited numerical precision. arXiv preprint arXiv :1608.06902, 2016.
>
> 2. Lu Hou, Quanming Yao, and James Tin Yau Kwok. Loss-aware binarization of
> deep networks. In 5th International Conference on Learning Representations, ICLR, 2017
>
> 3. Shu-Chang Zhou, Yu-Zhi Wang, He Wen, Qin-Yao He, and Yu-Heng Zou. Balanced quantization : An effective and efficient approach to quantized neural networks. Journal of Computer Science and Technology, 32 :667–682, 2017.
>
> 4. Peiqi Wang, Xinfeng Xie, Lei Deng, Guoqi Li, Dongsheng Wang, and Yuan Xie. Hit-net : Hybrid ternary recurrent neural network. Advances in Neural Information Processing Systems, 31, 2018.
>
> 5. Sheng Shen, Zhen Dong, Jiayu Ye, Linjian Ma, Zhewei Yao, Amir Gholami, Michael W Mahoney, and Kurt Keutzer. Q-bert : Hessian based ultra low precision quantization of bert. Proceedings of the AAAI Conference on Artificial Intelligence, 34(5) :8815–8821, 2020.
>
> 6. Tommaso Pacini, Emilio Rapuano, and Luca Fanucci. Fpg-ai : A technology-independent framework for the automation of cnn deployment on fpgas. IEEE Access, 11 :32759– 32775, 2023.
>
> 7. Jeffrey Chen, Sang-Woo Jun, Sehwan Hong, Warrick He, and Jinyeong Moon. Eciton :
> Very low-power recurrent neural network accelerator for real-time inference at the edge.
> ACM Transactions on Reconfigurable Technology and Systems, 17(1) :1–25, 2024.

---

> ### Author Response · Authors · 2024-11-24
> **Questions**
>
> 1. **Could you evaluate the performance of Block-HadamRNN and HadamRNN on edge devices? In particular, would you be able to provide the speedup and computational overhead that while using your techniques on these devices?**
>
> We addressed this critique in the previous post.
>
> 2. **Could you maybe think about making Tables 2 and 3 simpler to improve the readability of your results? Better visual perception may also result from the inclusion of a plot that contrasts the accuracy of all the suggested and benchmark approaches.**
>
> As described in the previous post, we include new plots in a new appendix.
>
> 3. **Could you maybe include a thorough example in the text that shows how a trained ORNN is subjected to the binarization or sparse ternarization algorithms? Could you also explain the Straight-Through Estimator's (STE) function in the parameterization process?**
>
> In practice, we do not binarize a pre-trained full-precision ORNN. Instead, as is standard in Quantization Aware Training (QAT), we directly optimize the quantized weights. The algorithm transitions from one quantized recurrent weight matrix to another, aiming to optimize the objective function. This process relies on the use of the Straight-Through Estimator.
>
> We present the Straight-Through Estimator for optimizing the binary vector $u\in\\{-1,1\\}^{d_h}$ appearing in (3) and (4) in a new appendix, Appendix C.1.
>
> Also, in Appendix C.2, we explicitly present the recurrent weight matrix for HadamRNN when $d_h = 4$.
>
> We hope the new appendix clarifies this point.

---

### Author Response · Authors · 2024-11-24
**Overall reply to the reviewers**

We would like to thank all the reviewers for their time spent reading the article, as well as for their positive comments and constructive criticism, which have greatly contributed to improving the quality of our work.

For each reviewer, we provide point-by-point responses to the issues raised in their report. The reviewers' comments are highlighted in bold, while our responses are in regular text. Below, we summarize the changes we will implement based on the reviewers' feedback.

The revised version of the article will be submitted by November 26.

Below, we summarize the modifications.

- Major modifications:
    - We provide a new appendix that describes the STE and presents an example of the recurrent weight matrix for $d_h=4$.
    - In addition to Tables 2 and 3, we illustrate the benefits of using HadamRNN and Block-HadamRNN through plots in a new appendix.
    - We provide an expanded bibliography on transformers that handle long-term dependencies. %and quantized transformers.

    - We include new benchmarks:
        - SST-2 and (if time permits) CoLA from GLUE.
        - HAR-2 and DSA-19, benchmarks for human activity recognition (from the FastGRNN article).
        - Time permitting, Yelp-5, a sentiment analysis benchmark (from the FastGRNN article).
    - We provide a comparison with binary transformers on the above GLUE benchmarks.
    - We provide the training times and a stability analysis.
- Minor modifications:
    - We list additional potential applications of Hadamard matrices in the conclusion.
    - We have proofread the article to eliminate any remaining typos.
    - We have labeled the axes of the plot in Figure 1.

---

### Author Response · Authors · 2024-11-27
**Revised paper submitted**

Dear reviewers and area chair,

We have uploaded the updated version of the article.

All changes are highlighted in red.

In comparison to the previously announced updates, we have made the following modifications:

1. The additional benchmarks are:
   - SST-2 and QQP from GLUE, with a comparison to all existing quantized BERT models addressing these benchmarks.
    - The IoT tasks HAR-2 and DSA-19 from FastGRNN.
2. We have included only the bibliography on quantized LLMs in Section 2, and we believe it is comprehensive. It is complemented by the comparison outlined above. The reasons for not including a bibliography on transformers handling long-term dependencies will be clear from the current content of Section 2.
3. There are a few modifications aimed at saving space. These are not highlighted in red and do not alter the meaning of the text.

All other modifications are as described in our previous posts.

Thank you very much for all your feedback. It has helped us improve our article.

Best regards,

The authors.

---

### Author Response · Authors · 2024-12-02
**Feedback**

Dear reviewers,
The end of the discussion period is approaching.
We would be very grateful for your feedback on the revised version.

Sincerely yours,
The authors.

---

> ### Comment · Reviewer_gcYV · 2024-12-02
>
> Dear Authors, after reviewing your work and thoroughly considering the changes you have made, I have followed up on my last comment and increased my decision score.

---

> ### Comment · Reviewer_wQjw · 2024-12-02
>
> Increased the score! All the best.

---

### Meta-Review · Area_Chair_PUrz · 2024-12-21

**Metareview:**

The paper presents a method to parameterize orthogonal RNNs with binary and sparse ternary weights using Hadamard matrices. This approach, including HadamRNN and Block-HadamRNN, resolves previous instability issues and achieves performance close to full-precision models. The reviewers are divided about this work. On one hand, this was not done previously, but on the other hand the experiments are weak with not diverse experiments and no clear path to real-world applicability. While the paper's outcome is less certain than other cases, it presents a promising new concept that warrants exposure to the ICLR audience. Therefore, I do recommend acceptance.

**Additional Comments On Reviewer Discussion:**

The reviewers raised various concerns regarding the real-world applications or the diversity of the datasets. Eventually, the authors made modifications on the paper, which improve the work.

---

### Decision · Program_Chairs · 2025-01-22

Accept (Poster)